# Disrupting abnormal neuronal oscillations with adaptive delayed feedback control

**Domingos Leite de Castro**[1,2], **Miguel Aroso**[1], **A Pedro Aguiar**[2], **David B Grayden**[3], **Paulo Aguiar**[1]*

[1]Neuroengineering and Computational Neuroscience Lab, i3S - Instituto de Investigação e Inovação em Saúde, Universidade do Porto, Porto, Portugal; [2]Faculdade de Engenharia, Universidade do Porto, Porto, Portugal; [3]Department of Biomedical Engineering, University of Melbourne, Melbourne, Australia

**Abstract** Closed-loop neuronal stimulation has a strong therapeutic potential for neurological disorders such as Parkinson's disease. However, at the moment, standard stimulation protocols rely on continuous open-loop stimulation and the design of adaptive controllers is an active field of research. Delayed feedback control (DFC), a popular method used to control chaotic systems, has been proposed as a closed-loop technique for desynchronisation of neuronal populations but, so far, was only tested in computational studies. We implement DFC for the first time in neuronal populations and access its efficacy in disrupting unwanted neuronal oscillations. To analyse in detail the performance of this activity control algorithm, we used specialised in vitro platforms with high spatiotemporal monitoring/stimulating capabilities. We show that the conventional DFC in fact worsens the neuronal population oscillatory behaviour, which was never reported before. Conversely, we present an improved control algorithm, adaptive DFC (aDFC), which monitors the ongoing oscillation periodicity and self-tunes accordingly. aDFC effectively disrupts collective neuronal oscillations restoring a more physiological state. Overall, these results support aDFC as a better candidate for therapeutic closed-loop brain stimulation.

*For correspondence:
pauloaguiar@i3s.up.pt

**Competing interest:** The authors declare that no competing interests exist.

## Editor's evaluation

Large populations of neurons are capable of entering pathological synchronous oscillations under a variety of conditions and work over many decades has found ways to disrupt such oscillations using stimulation in both open loop and closed loop configurations. This study adds useful results and methodology to this line of research, by providing solid evidence that delayed feedback control via electrical stimulation can, under certain conditions, terminate network level oscillations in cultured hippocampal neurons. The study provides analyses and simulation results that shed light on why some networks respond to such feedback control while others do not.

## Introduction

The excessive oscillatory synchronisation that emerges in neurological disorders, such as Parkinson's disease and epilepsy (**Uhlhaas and Singer, 2006**), can be disrupted using direct electrical stimulation (**Edwards et al., 2017**). Despite its potential, conventional brain stimulation, therapies still rely on continuous open-loop stimulation, leading to excessive stimulation and burdensome side-effects (**Kuo et al., 2018**). In theory, closed-loop stimulation guided by an efficient control algorithm should provide better outcomes, disrupting the pathological neuronal dynamics with minimal intervention. But, given the complexity of neuronal systems, it is very challenging to foresee what type of control algorithms would be effective. Multiple in silico studies focused on employing closed-loop control

methods to desynchronise pathological oscillations in computational models of neurological disorders (*Beverlin Ii and Netoff, 2012*; *Holt et al., 2016*; *Nabi et al., 2013*; *Weerasinghe et al., 2021*; *Witt et al., 2013*), but these have not been translated into studies with real neuronal systems to access its viability.

A particularly prevalent method in the field is delayed feedback control (DFC), a technique from chaos theory developed to stabilise/destabilise chaotic systems using minimal perturbations (*Pyragas, 1992*; *Rosenblum and Pikovsky, 2004a*). DFC controls the system by applying a feedback signal proportional to the difference between the current oscillation amplitude and the oscillation amplitude from a fixed delay into the past. DFC has been extensively applied in the context of in silico neuroscience, with studies focusing on specific disorders such as Parkinson's disease (*Rosenblum and Pikovsky, 2004b*; *Vlachos et al., 2016*; *Popovych et al., 2017*; *Daneshzand et al., 2018*; *Popovych and Tass, 2019*; *Yu et al., 2021*) and epilepsy (*Zhou et al., 2020*). However, there is still controversy in the literature regarding its efficacy and whether it may instead promote synchronisation, depending on the baseline levels of synchrony of the modelled network (*Dovzhenok et al., 2013*). Thus, it is critical to test the effectiveness of DFC with biological neuronal networks that display synchronised oscillatory firing. Neuronal cultures on microelectrode arrays (MEAs) are a useful testbed for this task: these cultures self-organise into complex interconnected networks with quasiperiodic synchronised bursting, and MEAs enable closed-loop modulation of the neuronal activity, providing concurrent recording and stimulation at multiple points of the population with high spatial and temporal resolution (*Obien et al., 2014*).

In this work, we implemented, for the first time, the DFC algorithm to disrupt the synchronised oscillatory bursting of biological neuronal populations. The cells were cultured on MEAs and the network activity was monitored and controlled in real time with electrical stimulation. We show that traditional DFC lacks the adaptability to cope with the malleable dynamics of biological networks and may actually impose a new firing periodicity. As an alternative, we developed and implemented an adaptive DFC (aDFC) algorithm that self-tunes to better predict the current oscillation phase and periodicity. We demonstrate that aDFC reduces the synchronisation of the network and efficiently disrupts the baseline oscillation. While doing so, we identified and characterised a new feature of biological neuronal networks: the intrinsic firing dynamics of a network unveils its propensity for neuromodulation. To better understand this relationship, we went back to the in silico neuronal networks and revealed that their controllability can be conditioned by specific parameters of the network architecture, such as the excitatory/inhibitory balance and overall synaptic weights of the network connections. Taking all together, we show that theoretical predictions regarding the efficacy of DFC might be misleading and we present an improved algorithm for closed-loop electrical stimulation. The knowledge gathered while testing these algorithms in vitro was relevant to recognise shortcomings of in silico simulations and may guide the development of better computational neuronal models. Our results revealed fundamental properties and considerations regarding the controllability of neuronal systems.

## Results

To assess the potential of DFC as a closed-loop brain stimulation protocol (*Figure 1A*, top) for suppressing pathological neuronal oscillations, we developed an in vitro setup composed of hippocampal networks cultured on MEAs, a workstation running the control algorithms and specialised hardware (2100MEA-System, Multichannel Systems [MCS]) interfacing both environments at very low latencies (*Figure 1A*, bottom). Our in vitro setup provides a powerful testbed to analyse DFC algorithms as it incorporates neuronal populations that display synchronous oscillations (here in the form of quasiperiodic network bursts [NB] with periodicities in the seconds range), an acquisition system that measures the population's extracellular activity at high spatiotemporal resolution enabling the inference of synchrony among neurons, on-demand electrical pulse generators to stimulate the network at high spatiotemporal resolution, and a feedback loop with low actuation latency in the milliseconds range (orders of magnitude below that of the neuronal dynamics under control).

The core of the DFC algorithm used (*Figure 1B*, black) is analogous to implementations reported in the literature (see Methods section for details). After spike detection (*Figure 1C*, top row), the instantaneous population firing rate is calculated (*Figure 1C*, middle row) and filtered by a damped oscillator (*Figure 1C*, bottom row in blue). The resulting signal is a good representation of the oscillatory

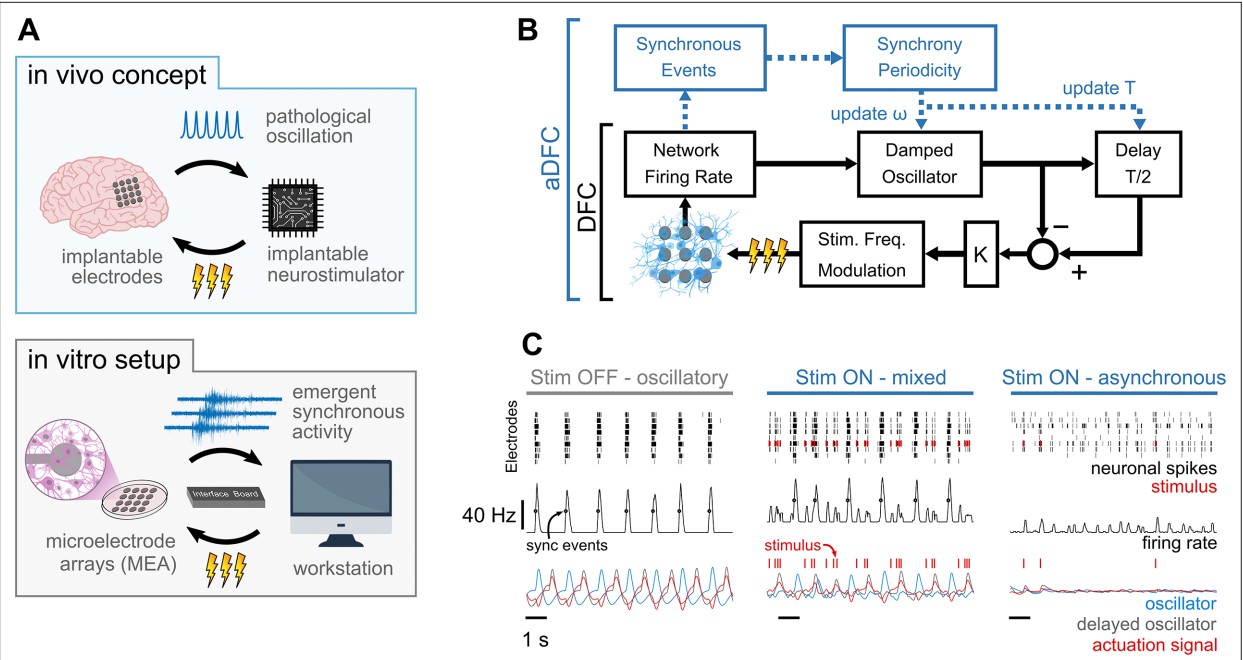

**Figure 1.** Controlling pathological oscillations with adaptive delayed feedback control (aDFC). (**A**) In vivo concept of implantable closed-loop neurostimulation and the analogous in vitro setup developed. Here, we have dissociated hippocampal neurons as a brain, microelectrode arrays as implantable electrodes, emergent periodic network bursting as pathological synchronous oscillations, a workstation as an implantable chip, and an external stimulator as an implantable neurostimulator. (**B**) Scheme of the DFC algorithms. The instantaneous population firing rate is calculated and filtered by a damped oscillator. The actuation signal (stimulation frequency) is obtained by subtracting the ongoing oscillation from the past half-cycle delayed oscillation. The natural frequency of the oscillator, $\omega$, and the duration of the half-cycle delay are updated online (adaptive component in blue) by detecting the synchronous events and their current periodicity, $T$. (**C**) Controller computations under different firing regimes – oscillatory (left), asynchronous (right), and mixed, i.e., oscillatory with induced sparse activity (middle). Spike detection is performed for the monitoring electrodes (top) and the spike train is convolved online with a square window to compute the instantaneous population firing rate (middle). Synchronous events (black circles) are detected using a threshold crossing method and used to update the oscillation periodicity. The actuation signal (red) is built by subtracting the oscillator and delayed oscillator signals (blue and grey, respectively) and translated to a stimulation frequency signal.

The online version of this article includes the following figure supplement(s) for figure 1:

**Figure supplement 1.** Representative example of adaptive delayed feedback control (aDFC) adaptability in a network with fast-changing dynamics.

behaviour of the network, such as that exhibited while the stimulation is turned OFF (*Figure 1C*, left). The feedback signal (*Figure 1C*, left, bottom in red) is then created by subtracting the ongoing oscillation from the half-cycle delayed oscillation (*Figure 1C*, bottom row in grey). This way, when the activity is oscillatory, the actuation signal is maximal at the antiphase of the neuronal oscillation (*Figure 1C*, left and middle). The obtained feedback signal is linearly translated to stimulation frequency by applying a fixed gain. Since the actuation signal is explicitly dependent on the neuronal oscillation, stimulation is highly activated when synchrony is high (*Figure 1C*, middle) and decreases as this firing pattern vanishes (*Figure 1C*, right). Expanding on the DFC algorithm, and to respond to the dynamic nature of biological neuronal systems, we improved canonical DFC to aDFC by adding an extra component that detects the synchronous firing events (*Figure 1C*, black circles in middle row). This way, aDFC monitors the current periodicity of the neuronal oscillation ($T$) and updates the delay duration ($T/2$) and natural frequency of the oscillator ($\omega$) accordingly (*Figure 1B*, blue and *Figure 1—figure supplement 1*).

The neuromodulation performance of the control algorithms was analysed in a series of experiments with multiple neuronal populations on MEAs. Each experiment had three stages: 5 min of spontaneous activity pre-stimulation (OFF), 5 min under stimulation (ON), and 5 min of spontaneous activity post-stimulation (OFF). Every network, here defined as a neuronal culture on a given day in vitro, was submitted to three different stimulation algorithms: standard DFC, aDFC, and random stimulation. During random stimulation, the electrical pulses were triggered by a Poisson process with an average frequency identical to that obtained in the preceding aDFC test. In all algorithms,

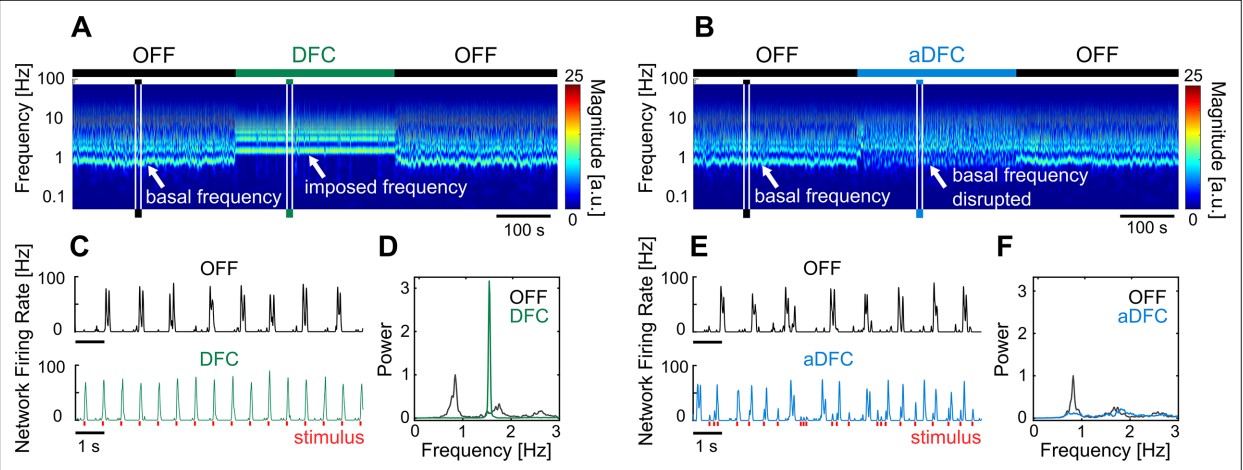

**Figure 2.** Representative example of the effect of delayed feedback control (DFC) and adaptive DFC (aDFC) stimulation protocols. (**A, B**) Wavelet transform of the instantaneous firing rate of a network under DFC (**A**) and aDFC (**B**) protocols. (**C, E**) Example of the instantaneous firing rate of the network before stimulation (black) and during DFC (**C**, green) and aDFC (**E**, blue) stimulation. The signals shown in C and E correspond to the narrow time windows marked in A and B, respectively. (**D, F**) Power spectrum density of the instantaneous firing rate of the network before stimulation (black) and during DFC (**D**, green) and aDFC (**F**, blue) stimulation.

stimulation was provided as monophasic negative voltage pulses at controlled timings. To ensure that the individual pulses had a comparable effect on the different neuronal networks, we chose a single stimulation electrode that triggered a neuronal response measured by neighbouring electrodes and the minimum voltage amplitude that achieved that response (see Methods section).

## Adaptive component is required due to the dynamic nature of neuronal populations

Changes in the neuronal dynamics caused by the stimulation protocols can be evaluated using the wavelet transform of the instantaneous network firing rate (*Figure 2A and B*). This representation evidences how the bursting periodicity of the network changes over time. Consistent inter-burst intervals (i.e. consistent neuronal oscillations) will appear as an horizontal line in the wavelet domain, corresponding to the fundamental frequency of the oscillation. When stimulation was activated, both DFC and aDFC showed clear effects on the neuronal dynamics. DFC stimulation imposed a change to a faster and very consistent oscillation (DFC periods in *Figure 2A and C*). This happens because DFC initially forces the network to fire in its antiphase, creating a change in the bursting periodicity. Since this change is not monitored to update the controller's delay, the delayed oscillator (*Figure 1C*, grey) no longer matches the antiphase of the ongoing oscillation. Therefore, stimulation becomes prejudicial as it is sent in phase with the new oscillation, reinforcing it (DFC period in *Figure 2C*).

On the other hand, aDFC disrupts the spontaneous oscillation frequency exhibited by the network, creating a firing regime where there is no predominant oscillation (aDFC periods in *Figure 2B and E*). In this case, the controller is constantly forcing the network to fire in antiphase and aDFC adapts as the network oscillation changes with the stimulation. To quantify this effect, we calculated the magnitude of the main oscillation by calculating the signal-to-noise ratio (SNR) of the power spectrum density (PSD) of the instantaneous firing rate for the ON and OFF segments (*Figure 2D and F*).

## Network controllability is conditioned by intrinsic population dynamics

The effect of the different stimulation protocols was evaluated using three metrics calculated for the OFF and ON periods: synchrony, firing rate, and oscillation intensity. Synchrony measures the degree of co-activation of multiple electrodes (see Methods); firing rate corresponds to the average number of spikes per electrode per unit of time (not to be confused with the instantaneous network firing rate signal presented in *Figure 2C and E*); oscillation intensity corresponds to the SNR of the PSD of the instantaneous firing rate signal (*Figure 2D and F*).

The three stimulation protocols were repeated multiple times for each network following the OFF-ON-OFF routine. Their effects on a given network are evidenced by normalising the values of each feature to the initial OFF period (before stimulation) across the multiple trials (*Figure 3A*). This also reveals how the dynamics are recovered once the stimulation is turned OFF. Note that, for the oscillation intensity, this normalisation corresponds to the difference between the ON and OFF periods, since the SNR scale is logarithmic [dB].

Some networks were consistently driven to distinct regions of the features space with each stimulation protocol (*Figure 3B*). We refer to these as *controllable* networks – hypothetically, an ideal controller could lead such networks to any desired reachable state. On the other hand, some networks did not exhibit consistent modulation when stimulated using the different protocols (*Figure 3C*). We termed these *uncontrollable* networks. Based on this definition, each network was classified as either controllable or uncontrollable by calculating the multivariate ANOVA (MANOVA) for the ON values of the three metrics (see Methods for details). Networks with low p-values ($p < 0.05$) (i.e. networks with significantly distinct neuromodulation results for the different stimulation protocols) were considered controllable (*Figure 3D*). Conceptually, controllability of a network is dependent on the assumed stimulation configuration: a network may be controllable with an array of stimulating electrodes, and uncontrollable using a single stimulating electrode.

Having recognised that some networks are controllable, we questioned if there are any common traits that make them more prone to neuromodulation. Even though there may be an indefinite number of hidden variables that influence the controllability of a neuronal population, some of these properties could be translated into specific firing dynamics, which we can access with the MEAs. If that is the case, then we expect the controllable and uncontrollable networks to have distinct fingerprints in their spontaneous firing dynamics. To evaluate this hypothesis, as well as ensure that the uncontrollable networks did not simply result from an ineffective stimulation electrode, we compared the spontaneous firing rate and synchrony during the initial OFF period (before stimulation) for all the experiments performed with all networks. This revealed that, as hypothesised, controllable and uncontrollable networks lie in different regions of the feature space (*Figure 3E* and *Figure 3—figure supplement 1*). Furthermore, controllable networks are in the centre, which is associated with intermediate levels of spontaneous firing rate and synchrony. This can be further evidenced by calculating the principal components (using principal components analysis) of the standardised map and projecting the values of each experiment along the principal component. A histogram shows that the experiments performed with controllable networks are in the centre, surrounded by a bimodal distribution of the experiments performed with uncontrollable networks (*Figure 3F*).

## aDFC disrupts oscillations and decreases overall synchrony in controllable networks

The performances of DFC, aDFC, and Poisson stimulation were evaluated by comparing their average effects on the three metrics across all the controllable network (*Figure 4*), where each datapoint corresponds to the metric's mean for all the experiments performed with a given network and stimulation protocol. Uncontrollable networks are not considered here as their results are, by definition, not reproducible. We included trials without stimulation for each network as a control group to capture the natural variability of each metric; in these, we calculated the metrics for two consecutive 5 min blocks of spontaneous activity, representing the initial OFF and ON periods.

The most significant distinction between control algorithms was evidenced at the level of the bursting oscillations (*Figure 4A*). Both aDFC and random stimulation led to a significant decrease in oscillation intensity (see also *Figure 3—figure supplement 2* for a representative wavelet transform under Poisson stimulation). The standard DFC algorithm, on the other hand, led to a significant increase in the oscillations. This increase in magnitude was followed by an increase in the oscillation frequency (*Figure 2A, C and D* and *Figure 3—figure supplement 3*). The only stimulation protocol that led to a significant decrease in synchrony was aDFC (*Figure 4B*) and its effect was significantly different from that of random stimulation. On the other hand, it was also the only method that significantly increased the average firing rate (*Figure 4D*), even though the average stimulation frequency was similar for the three methods (*Figure 4—figure supplement 1*).

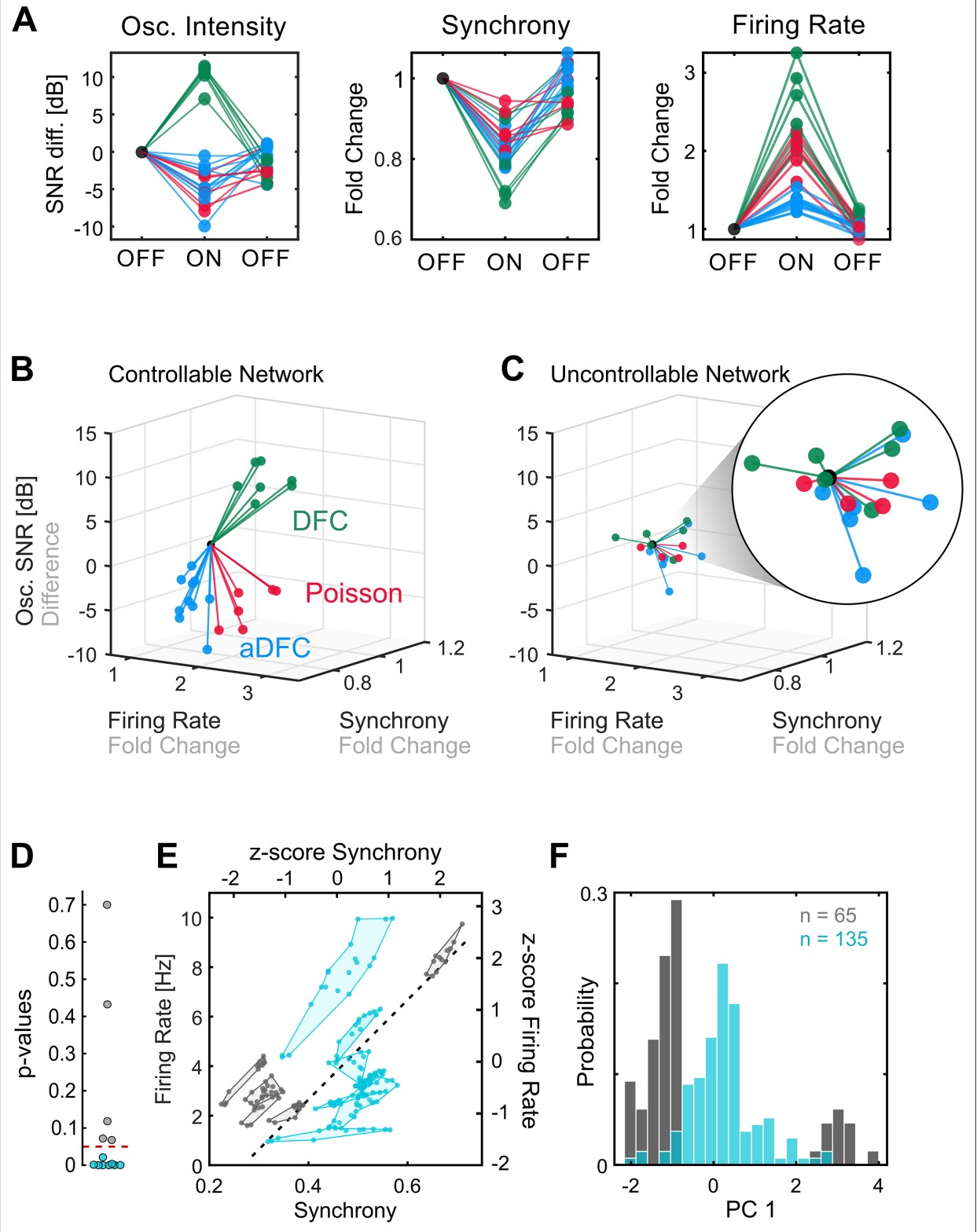

**Figure 3.** Comparison of controllable and uncontrollable networks. (**A**) Results obtained from multiple repetitions of the three stimulation protocols for a single network. We calculated the synchrony, firing rate, and oscillation intensity for the three segments – pre-stimulation (OFF), stimulation (ON), post-stimulation (OFF) – of each experiment and normalised each metric to the value of the pre-stimulation period. For the oscillation intensity, this normalisation corresponds to the difference and not ratio, since the signal-to-noise ratio (SNR) is given in dB (logarithmic scale). Each triplet of points corresponds to one experiment performed with either adaptive delayed feedback control (aDFC) (blue), DFC (green), or Poisson (red) stimulation.

*Figure 3 continued on next page*

*Figure 3 continued*

(**B**) Three-dimensional (3D) feature space – firing rate vs. synchrony vs. oscillation intensity – for the network in A. Each line corresponds to the metrics evolution from the initial OFF to ON (from the first to the second point in A). This network is considered controllable since each protocol consistently drove the network to a distinct subspace, meaning that it had a unique and reproducible effect. (**C**) Representative example of the 3D feature space for an uncontrollable network where the effect of each stimulation protocol is indistinguishable from the others. (**D**) Networks were separated into controllable (blue) and uncontrollable (grey) by computing the multivariate ANOVA (MANOVA) of the ON metrics for the different stimulation protocols. Networks with p-values lower than 0.05 had separable effects in the 3D feature space of the metrics and were thus considered controllable (8/13). (**E**) Pre-stimulation firing rate and synchrony for all experiments performed with each network, divided into controllable and uncontrollable networks. Each dot corresponds to one experiment performed with either DFC, aDFC, or Poisson in a given network. The experiments performed with the same network are confined in a polygon. The black traced line corresponds to the first principal component (PC1) of the standardised (z-score, i.e. mean equals zero and standard deviation equals one) synchrony and firing rate defining a relevant descriptor of the neuronal dynamics. (**F**) Distribution of the baseline neuronal dynamics along PC1, defined in E, showing that the controllable networks have an intermediate level of spontaneous firing rate and synchrony.

The online version of this article includes the following figure supplement(s) for figure 3:

**Figure supplement 1.** Controllable and uncontrollable networks across cultures, DIVs, and animal preparations.

**Figure supplement 2.** Representative example of the effect of Poisson stimulation in the frequency domain.

**Figure supplement 3.** Delayed feedback control (DFC) leads to an increase in oscillation intensity and frequency.

## Controllability is affected by synaptic strength and balance between excitation and inhibition

We then sought to understand whether the controllability subspace found for the neuronal cultures portrayed any fundamental property of these types of systems and what parameters could govern such behaviour. We modelled randomly connected networks of 1000 Izhikevich neurons (*Izhikevich, 2003*) with different proportions of regular spiking excitatory neurons and fast-spiking inhibitory neurons. We also varied the overall synaptic weight (see Methods for details). Each condition was simulated five times. The simulations were composed of two different periods: spontaneous activity (OFF) and controlled activity (aDFC ON). For each simulation, we computed the overall synchrony and firing rate for the OFF (*Figure 5A*) and aDFC ON periods (*Figure 5B*).

Low percentages of excitatory neurons and weak synaptic weights within the network led to sparse and chaotic activity (region 1 in *Figure 5A–C*). In this scenario, the controller is sporadically activated due to fluctuations of neuronal activity, creating brief moments of synchronisation (*Figure 5C*, top).

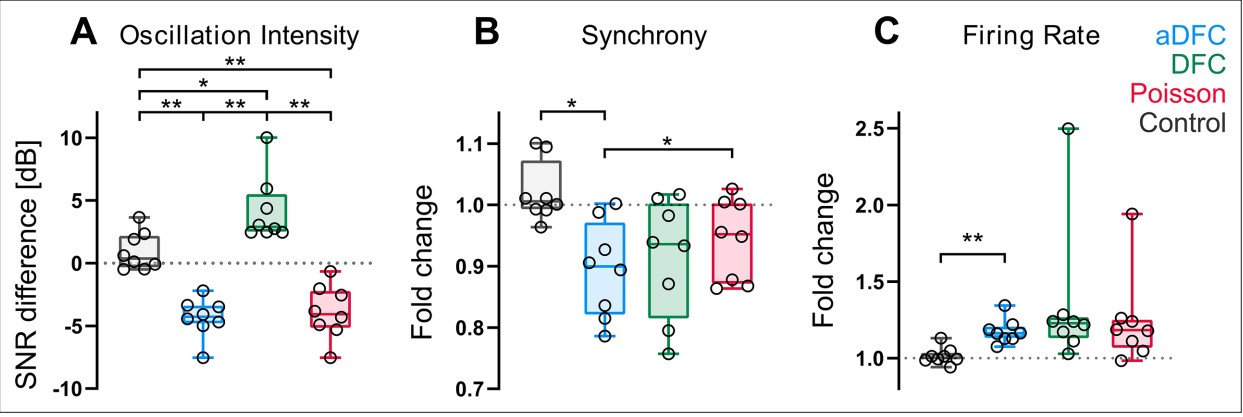

**Figure 4.** Effect of adaptive delayed feedback control (aDFC), DFC, and Poisson stimulation in controllable networks. Fold change in oscillation intensity (**A**), synchrony (**B**), and firing rate (**C**) of the 5 min stimulation period compared to the 5 min pre-stimulation period with aDFC (blue), DFC (green), and Poisson (red) stimulation. The control group (grey) corresponds to baseline recordings where there is no stimulation and thus portrays the natural variability of each metric within two subsequent 5 min blocks. Each point represents the average effect of a stimulation protocol across all trials for a given network. We used the one-way ANOVA with repeated measures as statistical test (*n*=8; *p<0.05; **p<0.01; details of statistical tests in *Supplementary file 1* [Table S1]).

The online version of this article includes the following figure supplement(s) for figure 4:

**Figure supplement 1.** Adaptive delayed feedback control (aDFC) allows the network to maintain some autonomous firing whereas DFC and Poisson tend to confine most network activity to stimulus responses.

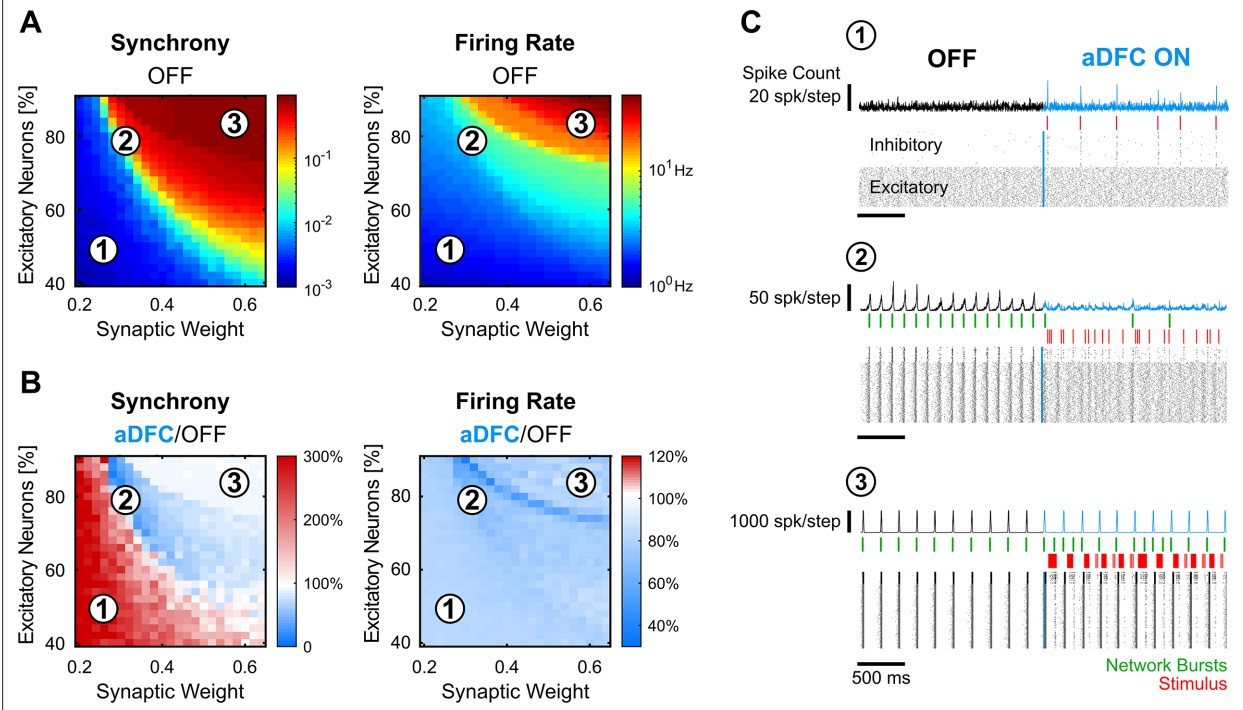

**Figure 5.** In silico networks also present a controllable subspace for intermediate baseline levels of firing rate and synchrony. (**A**) Level of synchrony (left) and firing rate (right) for the period of spontaneous activity. Each pixel represents the average synchrony and firing rate of five different simulations under the same conditions of excitatory/inhibitory balance and synaptic weight. (**B**) Average change in synchrony (left) and firing rate (right) of adaptive delayed feedback control (aDFC) ON period compared to the OFF period. (**C**) Representative example of the simulations under three distinct network parameterisations (1, 2, and 3 in A and B). Each panel has the instant spike count at the top, the raster plot at the bottom, and the network bursts (green) and triggered stimuli (red) in the middle.

The online version of this article includes the following figure supplement(s) for figure 5:

**Figure supplement 1.** The controllable region of in silico networks is also associated with intermediate levels of spontaneous firing rate and synchrony.

High percentages of excitatory neurons and strong synaptic weights led to highly ordered network dynamics with strong bursting activity (region 3 in *Figure 5A–C*). Here, the controller could not disrupt the dominating spontaneous activity (*Figure 5C*, bottom). However, in the transition between these two network states, there was a parameter subspace that generated intermediate baseline levels of firing rate and synchrony (region 2 in *Figure 5A–C*). In parallel to what was found in vitro, the networks were controllable in this intermediate region (see also *Figure 5—figure supplement 1*). Here, aDFC could desynchronise the network shortly after activating the stimulation (*Figure 5C*, middle). Thus, among the multiple possible parameters governing network controllability, the excitatory/inhibitory balance and overall network connectivity strength could explain the separation into controllable and uncontrollable networks seen in vitro.

## aDFC can promote transitions to stable asynchronous states

The majority of the in vitro cultures had monostable dynamics, presenting a very consistent firing pattern over time (e.g. OFF periods of *Figure 2*). In these networks, even though aDFC reduced neuronal synchronisation (*Figure 4C*) and disrupted baseline oscillatory activity (*Figure 2B, E and F* and *Figure 4A*), it generally did so by creating sparser bursting with no particular periodicity. In monostable networks, aDFC did not promote a phase transition to a stable asynchronous state (AS) as opposed to what is typically seen in DFC computational studies (*Rosenblum and Pikovsky, 2004a*; *Vlachos et al., 2016*; *Popovych et al., 2017*; *Daneshzand et al., 2018*; *Popovych and Tass, 2019*; *Yu et al., 2021*; *Zhou et al., 2020*).

We also found a less common type of networks that had multi-stable dynamics, spontaneously transitioning between different firing regimes. Multi-stable networks were particularly interesting when they presented sporadic transitions to an AS as it showed that these networks could autonomously

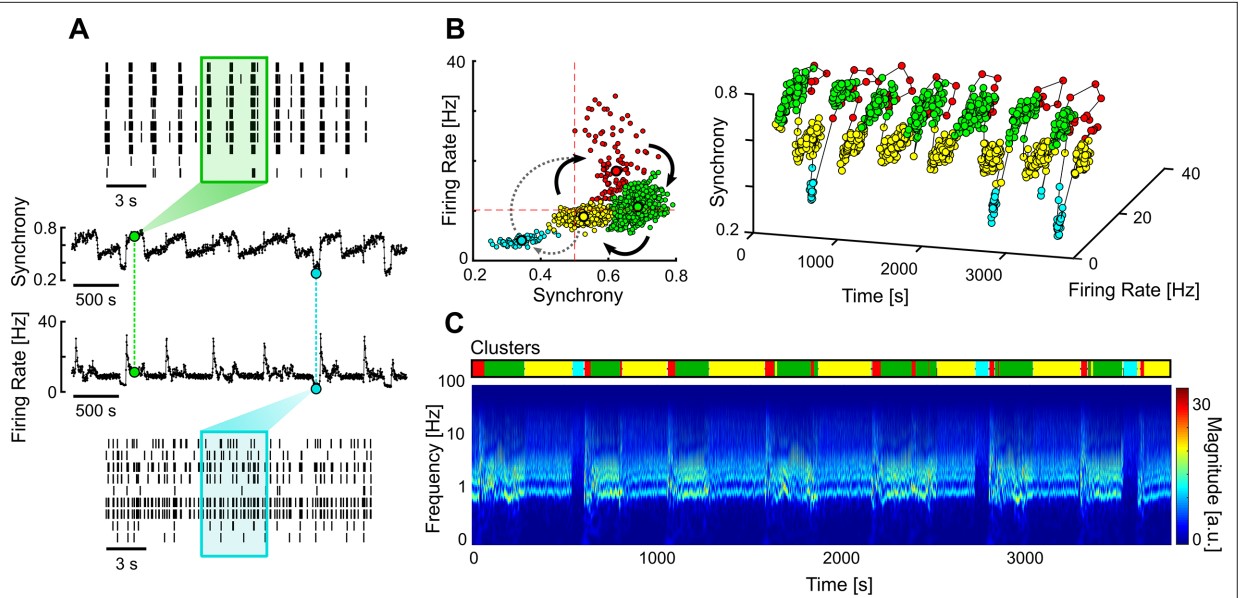

**Figure 6.** Multi-stable network with sporadic transitions to asynchronous state. (**A**) Spontaneous synchrony and firing rate over time (middle). Representative raster plots of the synchronous (top) and asynchronous states (bottom). The coloured boxes represent the sliding window over which the firing rate and synchrony are calculated. (**B**) Synchrony vs. firing rate phase plot evidencing the cyclic behaviour of the network, with sporadic transitions to the asynchronous state (blue cluster). The clusters were identified using unsupervised Gaussian mixture models. (**C**) Changes in the frequency domain correlate with the automatically identified clusters (top bar).

sustain asynchronous activity, even if only for short periods of time (*Figure 6*). The different stable states were identified using unsupervised clustering on the values of firing rate and synchrony calculated over a sliding window (*Figure 6A and B*). A given state was considered asynchronous if its average synchrony was below 0.5 and the mean firing rate was below the average of that network. The obtained clusters correlate with the different regimes seen in the frequency domain (*Figure 6C*).

To evaluate the effectiveness of each stimulation protocol in promoting transitions to AS, we compared the percentage of time spent in AS during the OFF and ON periods in multi-stable networks (*Figure 6*). Since our experimental protocol was composed of 5 min blocks (OFF-ON-OFF), the desynchronising effect of the closed-loop stimulation was only clear for networks that spontaneously exhibited rare and brief (considerably less than 5 min) AS. This way, a significant increase in the time spent in AS during the ON periods was most likely caused by the closed-loop stimulation. Out of all the networks tested, one presented the required criteria to evaluate this effect (*Figure 7A and B*).

aDFC consistently promoted desynchronisation in all the trials performed with this network, leading to a considerable increase of time spent in AS (*Figure 7C and D*). DFC stimulation, on the other hand, tended to impose a new bursting periodicity (as already reported), sometimes interrupted by transitions to AS (*Figure 7E and F*). However, the percentage of time spent in AS during DFC stimulation is within the range of spontaneous transitions and thus cannot be directly attributed to DFC. Random Poisson stimulation with identical stimulation frequency promoted sparser bursting with some transitions to AS, but these are also within the range of spontaneous transitions (*Figure 7G and H*). This means that the precisely timed stimuli determined by aDFC are causing the transitions to the AS. Also, the random Poisson stimulation, being an open-loop protocol, does not stop once the desired AS is achieved, which could be a significant drawback in a clinical setting.

## Discussion

Controlling brain activity is a long-standing goal in neuroengineering and holds great potential for multiple clinical applications. This is a challenging problem because neuronal systems have complex dynamics, virtually impossible to fully observe, and the space of actions available to an external controller is highly restricted. Most of these constraints are circumvented in computational studies, allowing researchers to explore solutions in a highly controlled environment. However, under such

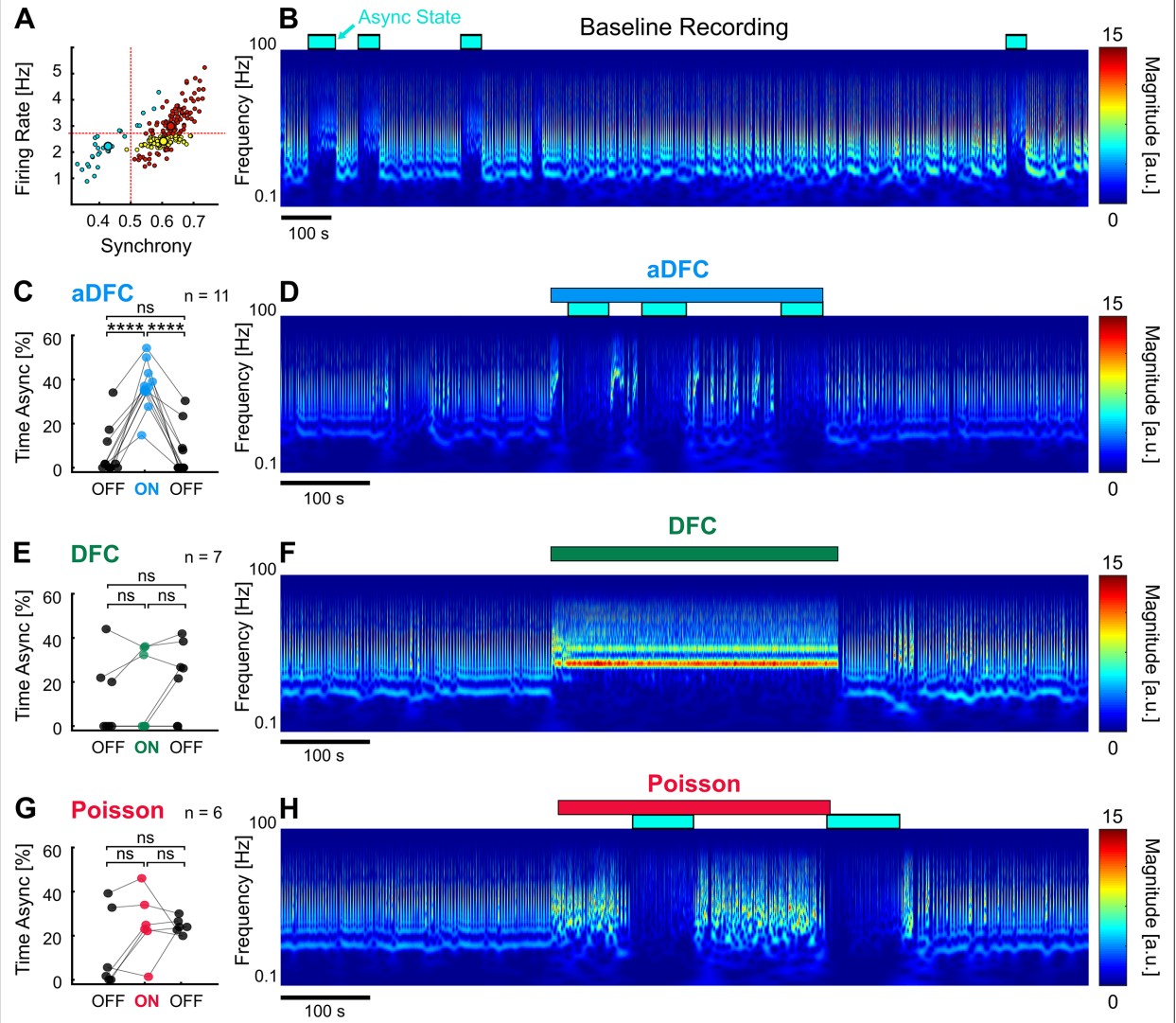

**Figure 7.** Adaptive delayed feedback control (aDFC) can promote transition to asynchronous state (AS). (**A**) States of spontaneous activity for a given multi-stable network. Blue cluster represents the sporadically reached AS. (**B**) Frequency domain of the recording considered in A. Blue rectangles represent the regions associated with the automatically identified AS. (**C, E, G**) Percentage of time in AS during the OFF-ON-OFF protocols for aDFC (**C**), DFC (**E**), and Poisson (**G**) stimulation. Each triplet of points represents an experiment performed with the corresponding stimulation protocol. We compared the time spent in AS during the ON and OFF periods using one-way ANOVA with repeated measures (****p<0.0001; details of statistical tests in *Supplementary file 2* [Table S2]). (**D, F, H**) Representative example of an experiment performed with aDFC (**D**), DFC (**F**), and Poisson (**H**) stimulation portrayed in the frequency domain. The stimulation period is represented with the associated coloured bar on the top. The identified AS are represented by the blue rectangles, as in B.

simplified conditions, these solutions may converge to algorithms that fail when applied to biological neuronal populations. It is, therefore, important to assess the performance of such algorithms with simple biological neuronal systems before moving to in vivo trials. In this work, we tested, for the first time, the efficacy of conventional DFC in an in vitro setting and reported that it promoted a new bursting frequency, instead of ablating it. This is in line with the claim that linear DFC could lead to synchronisation if the delay deviated substantially from the half-period (*Pyragas, 2006*). This was shown in silico by testing different delays with the same network. Here, however, the deviation from the half-period delay was caused by the fact that DFC stimulation induced a faster periodicity, while the delay was kept constant. The neuronal dynamics aligned with the non-adaptive controller to the point that almost all activity was confined to stimulus responses (*Figure 4—figure supplement 1*). This was not predicted by computational studies, possibly because the simple in silico networks used

were not as dynamic as biological ones and, therefore, could not sustain different synchronisation periodicities.

The adaptive version, aDFC, effectively disrupted the oscillations and decreased network synchrony. Interestingly, random stimulation, using similar stimulation frequency, was also able to achieve this, although without significantly decreasing synchronisation. Electrical stimulation may be acting as a disrupter of the network dynamics, as it has been proposed for high-frequency deep brain stimulation (*Chiken and Nambu, 2016*).

We showed that not all networks could be effectively modulated by each stimulation protocol. However, it is important to note that controllability is contingent on the neurostimulation conditions. In particular, the fact that we only used one stimulating electrode may restrict the pool of controllable networks. We did so to follow the principle that DFC should control the system using minimal intervention (*Rosenblum and Pikovsky, 2004a*). Also, despite the fact that aDFC decreased the synchrony of controllable networks, the neuronal activity was still mostly dominated by synchronous events. These results were far from those obtained in previous in silico studies, where the synchronous events vanish almost completely. We point to two possible causes. First, with our setup, we could only apply excitatory electrical stimulation, much like existing implantable stimulators. For this reason, when the actuation signal (here modulated as a stimulation frequency) was negative, we sent no input to the network. Nonetheless, in theory, using only the positive phase of the actuation signal should still prevent the formation of synchronous events, as it did in our in silico tests. So, it is possible that the limiting factor is the neuronal model itself. The unconstrained growth of our hippocampal cultures may have led to highly interconnected networks whose neurons simply cannot fire asynchronously. This is further supported by the fact that aDFC consistently promoted the AS in a network that could already sustain it autonomously. Again, these considerations are conditioned to the method used, whose main goal was to achieve desynchronisation with minimal intervention. Other methods can achieve network desynchronisation by following approaches that are more aggressive. Namely, it was already shown that continuous stimulation at higher frequencies (10–50 Hz), distributed across multiple electrodes, can prevent network synchronisation (*Wagenaar et al., 2005*). This is done, however, at the cost of highly increasing the overall network's firing rate, thus possibly establishing yet another non-physiological state. The goal of avoiding an abnormal prolonged increase in the population's firing rate is grounded on the understanding that excessive activity is also detrimental in a neuronal circuit. Whether aDFC is more efficient in disrupting pathological oscillations than high-frequency stimulation (HFS) used in current stimulation therapies remains unknown. Nevertheless, as opposed to fixed/continuous HFS, aDFC stimulation adapts to the population oscillation frequency and ceases once the network is desynchronised.

Despite being able to disrupt the baseline oscillation and decreasing synchrony levels, there is still margin to improve the aDFC algorithm by making it self-adaptive in terms of the actuation gain, pulse amplitude, and selection of stimulation electrode(s), all of which were tuned manually before starting the experiments. It remains to be explored whether such an adaptive controller would consistently achieve full desynchronisation in this type of networks.

To conclude, the technological advances made in implantable neurostimulators open the door for the use of algorithms capable of controlling pathological neuronal dynamics. We show that aDFC has the potential to be a promising candidate and believe that the considerations extracted from this in vitro study will support the translation to future in vivo experiments.

## Methods
### Cell cultures

The experiments followed both the European legislation regarding the use of animals for scientific purposes and the protocols approved by the ethical committee of i3S. The Animal Facility of i3S follows the FELASA guidelines and recommendations concerning laboratory animal welfare, complies with the European Guidelines (Directive 2010/63/EU) transposed to Portuguese legislation by Decreto-Lei no 113/2013, and is licensed by the Portuguese official veterinary department (DGAV, Ref 004461). Embryonic (E18) rat hippocampal neurons were cultured on six-well or nine-well chamber MEA (256-6well MEA200/30iR-ITO-rcr and 256-9wellMEA300/30iR-ITO, respectively) (MCS, Germany) with a density of $5\times10^4$ cells/well and $3\times10^4$ cells/well, respectively. Each well of the six-well MEA has 42 TiN

circular electrodes (array of 7×6), and the nine-well contains arrays of 26 TiN circular electrodes (6×5 without corners), each with a diameter of 30 μm. The experiments were performed between 13 and 55 days in vitro.

## Electrophysiological recordings

The recordings were performed with the MEA2100-256 System (MCS, Germany). The cell cultures were kept at 37°C and 5% $CO_2$ using a stage-top incubator (ibidi GmbH, Germany) adapted to the headstage of the MEA2100-256 System. The electrophysiology signals were sampled at 10 kHz and filtered with an HP at 200 Hz (second-order high-pass Butterworth filter). The neuronal spikes were detected and recorded using the Experimenter software from MCS. Peaks that were six STD (sometimes up to eight STD, depending on the background noise) below or above baseline were considered neuronal spikes. There was a deadtime of 3 ms after each detection to prevent biphasic spikes from being identified twice. This spike data was used for posterior offline analysis.

## Closed-loop computer application

The developed C# software was built using the McsUsbNet dynamic link library from MCS (https://github.com/multichannelsystems/McsUsbNet; *Multi Channel Systems MCS GmbH, 2023*). The software ran in parallel with the Experimenter software in a quad-core Intel i7-6700 with 32.0 GB RAM workstation. The raw neuronal stream of data was received in batches at 10 kHz. Each batch had one sample point per electrode and was transferred directly from MEA2100-256 System via USB 3.0 high-speed cable. The signal processing was analogous to that running on Experimenter: the raw data was high-pass filtered at 200 Hz and the spike detection was performed using the same algorithm (see Electrophysiology recordings section). The C# application had access to the MEA2100-256 System stimulators via USB 3.0, allowing us to trigger stimulation by software. The app also commanded the beginning of the recordings in Experimenter. The overall closed-loop latency associated with the activation of the stimulus was below 40 ms. The application can be freely available for download in GitHub (https://github.com/NCN-Lab/aDFC, copy archived at *Leite de Castro and Neuroengineering and Computational Neuroscience Lab, 2024*, GNU General Public License v3.0).

## DFC algorithm

The C# application ran the DFC algorithm online. The application only monitored electrodes that had an average firing rate of above 0.1 Hz, measured before starting the experiments. The instantaneous firing rate (*FR*) of an active electrode was calculated in a square moving window of 0.1 s (sometimes tuned to 0.2 or 0.5 before starting the experiments depending on the dynamics of the network). The *FR* signal corresponds to the number of spikes per active electrode per unit of time. The *FR* of the monitored network was filtered using a damped oscillator (*Popovych et al., 2017*; *Daneshzand et al., 2018*),

$$\ddot{x} + \omega\dot{x} + \omega^2 x = kFR\left(t\right) \tag{1}$$

where $\omega$ denotes the natural frequency of the oscillator and $k$ is a scaling coefficient. Here, we considered $k = \omega$, which prevents the oscillator amplitude from scaling with the different ranges of periodicities exhibited by different cultures. We used $\dot{x}$ as the output of *Equation 1*. The stimulation frequency (*SF*) is obtained by subtracting the current oscillation from the half-period shifted oscillation,

$$SF\left(t\right) = K\left(\dot{x}\left(t - \frac{T}{2}\right) - \dot{x}\left(t\right)\right) \tag{2}$$

where $K$ represents the stimulation gain and $T$ is the oscillation period. A stimulus is triggered when

$$t > t_{laststim} + \frac{1}{SF\left(t\right)} \tag{3}$$

where $t$ is the current time and $t_{last\ stim}$ is the time stamp of the last stimulus. *Equation 3* was bounded so that the stimulus was only sent when $1 < SF(t) < 20$. The upper bound is required due to the latency in the stimulus activation (below 40 ms) and the lower bound guarantees that low background values of $SF$ do not trigger stimulation (and that there is no negative stimulation frequency). Here, $T$ and $\omega$ are adapted based on the real-time detection of synchronous events. The synchronous events (NB) were detected when the $FR$ signal was above a certain threshold, typically 10 Hz, considering a minimum interval between NB of 0.1 s. The threshold and minimum NB interval were sometimes adjusted before starting the experiments to better suit the observed data.

## Experimental protocol

Before starting the experiments, we monitored the networks for up to 10 min to identify the active electrodes. The application calculates the average firing rate for all the electrodes and selects those firing above 0.1 Hz. During this preliminary period, we also tuned the parameters for the detection of synchronous events, if necessary.

Each stimulus unit was a negative electrical pulse with 200 μs duration. The choice of stimulation electrode and pulse amplitude was based on the premise that the stimulus should promote a response in the neighbouring electrodes using minimal stimulation. We probed different electrodes manually using Experimenter software, starting with the most active ones. For each electrode, we tested different stimulus amplitudes from –400 to –1000 mV in steps of –100 mV. When a given electrode and stimulation amplitude evoked a consistent response in nearby neurons, we fixed the stimulus amplitude to 120% of that value to account for eventual adaptations of the network, which could render the stimulation ineffective.

The experiments started with a recording of 15 min of spontaneous activity. The choice of stimulation protocols – DFC, aDFC, and Poisson – for the following trials was randomised. Each trial had three periods: 5 min without stimulation (OFF), 5 min with stimulation (ON), and 5 min without stimulation again (OFF). For the DFC and aDFC protocols, the stimulation followed the $SF(t)$ signal determined by each algorithm. For the Poisson protocol, the stimulation followed a Poisson distribution of stimuli with an average frequency equivalent to that used in the previous aDFC trial. The time interval between consecutive trials was in the order of 5 min. A precise time interval between trials was not required/imposed as, once stimulation stopped, the firing dynamics returned to the baseline regime very quickly (*Figure 2A and B* and *Figure 3A*).

## Selection criteria

We only performed experiments on networks that displayed consistent oscillatory bursting activity with periodicities between 0.5 and 5 s, approximately. Some cultures presented the required dynamics but were rejected because they were (or became) irresponsive to stimulation. In order to have statistical robustness in the evaluation of the different stimulation protocols, we only analysed networks that were submitted to at least three trials with each protocol. A total of 14 networks fulfilled the criteria to proceed with the experiments and analysis. Here, we define a network as being a neuronal culture in a given day in vitro. We did this because the neuronal dynamics of the same culture of neurons may change drastically from day to day, so we considered it a different dynamical system. The 14 networks came from 11 independent cultures, from a total of five different animal preparations (*Figure 3—figure supplement 1*).

## Signal processing
### Features of neuronal activity

The offline analysis of neuronal data was based on the spikes recorded with Experimenter. We used three features to characterise the activity of the different networks: average firing rate, synchrony, and the oscillation intensity. The average firing rate corresponds to the average number of spikes per electrode per unit of time. The synchrony measure is based on the $\chi$ metric, which compares the variance of the network activity with the variance of each neuron's activity (*Golomb and Rinzel, 1994*),

$$\chi^2 = \frac{Var\left(\frac{1}{N}\sum_{i=1}^{N} V_i\left(t\right)\right)}{\frac{1}{N}\sum_{i=1}^{N} Var\left(V_i\left(t\right)\right)},\tag{4}$$

where $N$ is typically the number of neurons, but here corresponds to the number of electrodes, and $V_i$ $(t)$ is typically the voltage of neuron $i$, but here corresponds to the spike train of electrode $i$ convolved with a square wave of 50 ms and binarised. This signal defines the periods on which a given electrode is considered active. We are thus measuring the degree of co-activation of the multiple electrodes, regardless of the number spikes exhibited by different electrodes at the moment of synchronisation. We did not want this factor to leak to the synchrony measure because different electrodes may be synchronised but with different number of spikes per activation (which could be due to different number of neurons surrounding each electrode, for example).

The oscillation intensity is measured by calculating SNR of fundamental frequency of the PSD. This was calculated using the MATLAB functions *pwelch* for the PSD and *snr* for the SNR, which computes the ratio of the summed squared magnitude of the fundamental frequency to that of the noise. The *snr* function automatically detects the fundamental frequency and assures that the harmonics and DC component are not mistaken as noise. The fundamental frequency corresponds to the frequency at the peak of the PSD.

For the firing rate and synchrony, the effect of each stimulation protocol was evaluated by calculating the metric's fold change from OFF to ON. For the oscillation intensity we calculated the difference between the OFF and ON periods, since it is measured in dB, which is a logarithmic scale.

## Unsupervised clustering of neuronal dynamics

The different states of neuronal activity of a given recording (**Figure 6**) were identified by calculating the firing rate and synchrony in a moving window. The width of the moving window corresponded to the average duration of five periodic events in the analysed recording. The moving step was half the window width. The clusters of neuronal dynamics were automatically identified with Gaussian mixture models (GMMs) applied to the bidimensional matrix of firing rate and synchrony. We tested GMMs with one to four groups and each condition was repeated 10 times with different initial values. The ideal clustering was chosen based on the Bayesian informative criterion. We calculated the centroid of each cluster and considered the AS as being those with mean synchrony below 0.5 and mean firing rate below the average for that network.

## Statistical analysis

### Controllable and uncontrollable networks

The classification of networks as controllable or uncontrollable (**Figure 3**) was based on the assumption that controllable networks could be driven to specific subspaces of neuronal activity with a given stimulation protocol whereas the uncontrollable networks could not. We started by calculating the fold change of the four metrics (synchrony, firing rate, oscillation frequency, and intensity) for all trials performed with each network. To evaluate whether the spaces reached during stimulation were statistically different, we calculated the MANOVA considering the fold change of the three metrics, simultaneously. Here, the null hypothesis was that the modulation results were identical for all stimulation protocols.

### Neuromodulation results for controllable networks

For a given network, we calculated the average fold change in synchrony and firing rate, and the difference in SNR of the main oscillation across all the trials performed with each stimulation protocol. Then, the comparisons between groups were performed using one-way ANOVA with repeated measures.

### Time spent in AS

To compare the efficacy of the different stimulation protocols in promoting the transition to a stable AS in a multi-stable network, we calculated the percentage of time spent in the AS during the OFF and ON periods for all the trials. We then compared if there was a significant change from OFF

pre-stimulation to ON, from ON to OFF post-stimulation and from OFF pre-stimulation to OFF post-stimulation using one-way ANOVA with repeated measures.

### In silico model

Our in silico networks were composed of 1000 randomly connected Izhikevich neurons with varying percentages of regular spiking excitatory neurons ($a$=0.02; $b$=0.2; $c$ = –65; $d$=8) and fast-spiking inhibitory neurons ($a$=0.1; $b$=0.2; $c$ = –65; $d$=2) (*Izhikevich, 2003*). The neurons were connected in an all-to-all architecture of varying synaptic weights according to the connectivity matrix $S$. The overall synaptic weights of the different simulations were modulated by scaling $S$. Each condition was simulated five times. The model was based on the script provided in https://www.izhikevich.org/publications/net.m.

The simulations had an initial period of 500 ms for the network activity to stabilise. After the stabilisation period, we had 2000 ms of spontaneous activity and 2000 ms under the aDFC stimulation protocol. The aDFC algorithm was identical to that applied in vitro, considering only the positive component of the actuation signal. A pool of 100 neurons was randomly selected for stimulation. Each electrical pulse corresponded to an instantaneous input of 20 mA to the stimulated neurons. The firing rate and synchrony measures were computed using the same algorithms as those used in vitro (see Signal processing section) and averaged across the five repetitions of each simulation condition.

## Acknowledgements

This work was supported by la Caixa Foundation, in the scope of the grant CaixaResearch Health 2022 HR22-00189, and by Prémio Mantero Belard, Santa Casa da Misericordia de Lisboa MB-12-2022. Domingos Castro was funded by FCT – Fundação para a Ciência e a Tecnologia, grant contract SFRH/BD/143956/2019.

## Additional information

### Funding

| Funder | Grant reference number | Author |
| --- | --- | --- |
| La Caixa Foundation | HR22-00189 | Paulo Aguiar |
| Santa Casa da Misericórdia de Lisboa | MB-12-2022 | Paulo Aguiar |
| Fundação para a Ciência e a Tecnologia | SFRH/BD/143956/2019 | Domingos Leite de Castro |
| Fundação para a Ciência e a Tecnologia | CEECIND/03415/2017/CP1392/CT0003 | Miguel Aroso |

The funders had no role in study design, data collection and interpretation, or the decision to submit the work for publication.

### Author contributions

Domingos Leite de Castro, Conceptualization, Software, Formal analysis, Investigation, Visualization, Methodology, Writing – original draft, Writing – review and editing; Miguel Aroso, Investigation, Methodology, Writing – review and editing; A Pedro Aguiar, David B Grayden, Methodology, Writing – review and editing; Paulo Aguiar, Conceptualization, Resources, Supervision, Funding acquisition, Investigation, Methodology, Project administration, Writing – review and editing

### Author ORCIDs

Domingos Leite de Castro (ID) https://orcid.org/0000-0002-2539-0311
Miguel Aroso (ID) http://orcid.org/0000-0002-3118-0185
Paulo Aguiar (ID) http://orcid.org/0000-0003-4164-5713

### Ethics

The experiments followed both the European legislation regarding the use of animals for scientific purposes and the protocols approved by the ethical committee of i3S. The Animal Facility of i3S follows the FELASA guidelines and recommendations concerning laboratory animal welfare, complies with the European Guidelines (Directive 2010/63/EU) transposed to Portuguese legislation by Decreto-Lei no 113/2013 and is licensed by the Portuguese official veterinary department (DGAV, Ref 004461).

### Decision letter and Author response

Decision letter https://doi.org/10.7554/eLife.89151.sa1
Author response https://doi.org/10.7554/eLife.89151.sa2

## Additional files

### Supplementary files

• Supplementary file 1. Details of the statistical tests used in *Figure 4* to compare the modulation results of the different stimulation protocols in controllable networks. We used repeated measures one-way ANOVA with multiple comparisons.

• Supplementary file 2. Details of the statistical tests used in *Figure 7* to compare percentage of time spent in asynchronous state for each stimulation protocol in a multi-stable network. We used repeated measures one-way ANOVA with multiple comparisons.

• Supplementary file 3. Details of the statistical tests used in *Figure 3—figure supplement 2* to compare fraction of spontaneous spikes during stimulation for the three stimulation protocols. We used repeated measures one-way ANOVA with multiple comparisons.

• MDAR checklist

### Data availability

The data that support the findings of this study are openly available in ZENODO at the following URL/DOI: https://doi.org/10.5281/zenodo.10138445.

The following dataset was generated:

| Author(s) | Year | Dataset title | Dataset URL | Database and Identifier |
|---|---|---|---|---|
| Leite de Castro D, Aguiar P | 2020 | Disrupting abnormal neuronal oscillations with adaptive delayed feedback control | https://doi.org/10.5281/zenodo.10138445 | Zenodo, 10.5281/zenodo.10138445. |

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
