## [Editor Report]

Large populations of neurons are capable of entering pathological synchronous oscillations under a variety of conditions and work over many decades has found ways to disrupt such oscillations using stimulation in both open loop and closed loop configurations. This study adds useful results and methodology to this line of research, by providing solid evidence that delayed feedback control via electrical stimulation can, under certain conditions, terminate network level oscillations in cultured hippocampal neurons. The study provides analyses and simulation results that shed light on why some networks respond to such feedback control while others do not.

---

## [Decision Letter]

**Decision letter after peer review:**

Thank you for submitting your article "Disrupting abnormal neuronal oscillations with adaptive delayed feedback control" for consideration by *eLife*. Your article has been reviewed by 2 peer reviewers, and the evaluation has been overseen by a Reviewing Editor and Panayiota Poirazi as the Senior Editor.

*Reviewer #1 (Recommendations for the authors):*

Suggestions for improved or additional experiments, data or analyses, especially those directed at increasing the impact of the work and making it suitable for *eLife*:

Page 4:

In the text, the Figure 1C middle part is never mentioned, indeed the "mixed" is also not explained and its meaning is not fully clear.

"In all algorithms, stimulation was provided as monophasic negative voltage pulses at controlled timings."

Why was this type of stimulation chosen? Was it based on current literature? What is the advantage of this type of stimulation vs a biphasic pulse?

Section 2.1:

It was not clear to me the advantage of representing the neuronal dynamics using the wavelet transform of the firing rate, since in the rest of the article this is represented by the DFT. Could you elucidate this choice?

"Consistent inter-burst intervals will appear as a low frequency horizontal line" – Why? If the ISI is constant but the firing rate is high, shouldn't it appear as a higher frequency horizontal line?

Figure 2:

I missed seeing the same analysis performed in the random stimulation condition – it could be included as a supporting figure just for completeness.

Section 2.2:

The definition of firing rate should appear before, since it is one of the main variables used in the previous section.

Defining the oscillation frequency and intensity based on the main peak does not seem to make sense for the aDFC condition. Your premise is that this treatment disrupts periodicity (which it appears to do) then, in theory, there should not be a defined "peak" in the frequency domain. This identified peak will rest within the noise levels, rendering it "random". Maybe a threshold could be set, i.e., if the signal-to-noise ratio of a peak is X, then it is considered, otherwise the frequency and intensity of the oscillation are respectively none and zero. Furthermore, the "randomness" of this oscillation frequency can be seen in Figure 3, where the trials with aDFC show a very large range of main frequencies.

Figure 3:

How is the z-score calculated?

Section 2.3:

Figure 4A show oscillation intensity yet, in the main text, you claim that "standard DFC led to more consistent oscillations". How does intensity relate to consistency? DFC shows a higher inter-trial variance than the other conditions (less consistent?)

Figure 4:

All variables are studied in four different conditions, yet the authors use t-tests to evaluate statistical significance. This is the wrong statistical test to use – an ANOVA should be used. In this situation, multiple t-tests will lead to an increase in false positives.

Section 2.4:

Did you quantify the excitatory/inhibitory balance in your MEA cultures? Could be interesting to compare to the simulation results.

Section 2.5:

The authors state that multiple trials were conducted in the same network but do not mention how long the network was allowed to recover in between trials, which is necessary information.

Furthermore, the authors state that only one network showed this effect (spontaneous AS time prior to stimulation). If only one network showed this among 14, how biologically relevant is this? How can you be sure that this is not a consequence of a confounding variable (not properly developed network, etc). How old was this network? Was it also used in other DIVs?

Figure 7:

The effect of the Poisson stimulation appears to be more similar to aDFC than random. A more in depth discussion would be helpful to better understand how a "random" stimulation could cause such a "block-type" effect.

The comparison between OFF-ON-OFF states refers to a comparison of the same object in different timepoints. Thus, t-tests are not the correct statistical test here, but rather a repeated measures ANOVA (or non-parametric equivalent) must be used.

Methods:

The authors state that experiments were performed between 13 and 55 DIV. When considering cultured neuron networks, this range makes a gigantic difference in their behavior, since the culture will not be fully mature at DIV 13, and the neuronal health at DIV 55 will already have declined. Only cultures at equivalent maturity levels should be compared.

Authors define "network as a neuronal culture in a given day in vitro". This needs to be clarified for statistical analysis. With this definition, does it mean that if two experiments are performed in the same neuronal culture but in two different days, they are considered two independent networks? While the network behavior might change throughout days in vitro (and network maturity), it is not possible to state that the behavior of the network is independent from its previous state. Extra care should be had when performing statistical analysis on those data – they should be handled as the same "object" at different timepoints, NOT independent networks.

The method used to determine the main frequency and intensity of the oscillation ASSUMES that there is a peak frequency, which may not be true and lead to erroneous results and conclusions.

*Reviewer #2 (Recommendations for the authors):*

This was a clear, well-written study. I only have a few suggestions/comments to the authors:

0) Can you discuss whether you expect brain networks to operate in uncontrollable vs. controllable regimes? Is there evidence that brain networks should be controllable?

1) Can you establish a correspondence between the controllable subspace in the in silico model in Figure 5 and the controllable network properties in Figure 3? For example, can you make Figure 5B left (change in synchrony as a function of synaptic weight vs. excitatory neurons) into a synchrony as a function of firing rate vs. synchrony plot? Then superimpose data from Figure 3E so that one can see whether the in silico network properties are similarly predictive of controllable vs. uncontrollable networks as the in vitro network properties. If there isn't a clear correspondence between the in silico and in vitro networks, please discuss.

2) Could you either provide experimental data or a discussion of previous studies that have tested conventional high frequency stimulation (~100-120 Hz) and studied its effects on neural synchronization? How might you expect your aDFC results to compare to conventional high frequency stimulation? There is some mention in the discussion of a study that uses 10-50 Hz and is able to prevent synchronous states from forming at the cost of a high firing rate. Why is the high firing rate a problem? Might this be the source of undesirable side effects of stimulation? Please discuss this in more detail in the discussion as it directly sheds light on how useful your algorithm may be compared to conventional stimulation.

3) Please discuss in the discussion the limits of the in vitro model, particularly in reference to the low frequency synchrony that the populations exhibit (~1 hz) compared to the pathological oscillatory activity thought to be relevant to PD (15-30 hz) or ET (5-7 Hz). Do you expect your algorithm to have difficulties suppressing these higher frequency oscillations ? Are there in vitro models that exhibit higher frequency oscillations that could be used to test this algorithm?

4) Can the authors comment on how many units must be monitored from in order to make an accurate estimate of the underlying neural population's oscillatory frequency? Can LFP features be used instead of multiple single neuron recordings? This has a direct bearing on how readily translatable this technology is to current neurostimulation devices.

5) During aDFC how often were the frequency (w) and period time (T) updated?

6) Where was the ground for the stimulation electrode? Does your stimulation better mimic bipolar or monopolar stimulation in current therapeutic DBS therapies? What was the impedance of the stimulation electrode? Some neuromodulation fields use current (not volts) to report their stimulation amplitude and it would be helpful for those building on this study to have both quantities reported. In addition, please report the electrode surface area so that current density calculations can be made.

7) Why was stimulation frequency the metric that was updated by DFC and aDFC as opposed to stimulation amplitude?

---

## [Author Response]

Reviewer #1 (Recommendations for the authors):Suggestions for improved or additional experiments, data or analyses, especially those directed at increasing the impact of the work and making it suitable for eLife:Page 4:In the text, the Figure 1C middle part is never mentioned, indeed the "mixed" is also not explained and its meaning is not fully clear.

The middle part depicts the initial instants after stimulation is turned ON. There is still a strong oscillatory component, interleaved with invoked activity, allowing the reader to understand the concept that DFC tends to force the network to fire out of phase. We used the term “mixed” because the activity is still oscillatory, but with moments of sparser activity. This middle part also helps the reader understand that the stimulation is highly activated when the synchrony is high, and vanishes as the synchrony disappears, as shown on the image on the right.

Added to the manuscript:

Page 4: “This way, when the activity is oscillatory, the actuation signal is maximal at the antiphase of the neuronal oscillation (Figure 1.C, left and middle). The obtained feedback signal is linearly translated to stimulation frequency by applying a fixed gain. Since the actuation signal is explicitly dependent on the neuronal oscillation, stimulation is highly activated when synchrony is high (Figure 1.C, middle) and decreases as this firing pattern vanishes (Figure 1.C, right).”

Legend of Figure 1. “(C) Controller computations under different firing regimes – oscillatory (left), asynchronous (right) and mixed, i.e., oscillatory with induced sparse activity (middle).”

"In all algorithms, stimulation was provided as monophasic negative voltage pulses at controlled timings."Why was this type of stimulation chosen? Was it based on current literature? What is the advantage of this type of stimulation vs a biphasic pulse?

When using voltage pulses, the advice from the manufacturer (Multichannel Systems) for titanium nitrite (TiN) electrodes (as in our MEAs) is to use monophasic negative pulses to assure charge balance at the electrode surface: *“When using TiN electrodes, it is extremely important to not charge the electrodes positively, as this will lead to electrolysis. (…). Therefore, when using voltage driven stimulation, it is important to apply negative voltages only. Positive voltages will shortly charge the electrodes positively, even though the electrode is discharged at the end of the pulse. As a consequence, biphasic voltage driven stimulation is not recommended.”*, MEA2100-System User Manual, page 49

https://www.multichannelsystems.com/sites/multichannelsystems.com/files/documents/manuals/MCS_MEA2100-System_Manual.pdfSection 2.1:It was not clear to me the advantage of representing the neuronal dynamics using the wavelet transform of the firing rate, since in the rest of the article this is represented by the DFT. Could you elucidate this choice?

The wavelet transform allows the reader to interpret visually the changes in frequency components across time, which is not possible with the Fourier transform. With the wavelet transform, it is clear that DFC and aDFC produce very different modulation outcomes as soon as the stimulation is turned ON, and that the spontaneous activity is quickly recovered when stimulation is turned OFF. In addition, introducing this concept makes the interpretation of results compiled in Figures 6 and 7 much more intuitive. In these figures, the wavelet transform is crucial to reveal how the different firing regimes change across time. Finally, we believe that this type of representation – the wavelet transform of the instantaneous firing rate of the entire population – is a very useful technique to analyse the population dynamics in long-term recordings and is not yet rightfully explored in the in vitro MEA literature. The raster plot typically used to represent spike data of neuronal populations is only useful in the seconds range; with the wavelet transform, we can analyse the populations’ dynamics across minutes and even hours.

"Consistent inter-burst intervals will appear as a low frequency horizontal line" – Why? If the ISI is constant but the firing rate is high, shouldn't it appear as a higher frequency horizontal line?

In the manuscript, we use two complementary by distinct metrics for the firing rate: the network instantaneous firing rate (the signals shown in Figure 2.C and E) and the average firing rate (shown in Figure 3). The network instantaneous firing rate is a time series that shows the number of spikes occurring in the network per electrode at each instant in time. The average firing rate is a scalar that represents the average number of spikes per electrode per second in the considered 5 min block (ON or OFF).

The wavelet transform is calculated based the network instantaneous firing rate. If there is a consistent inter-burst interval (let us say 1 second interval, for example) that means that there is a well-defined fundamental frequency (1 Hz, in this example) in the instantaneous firing rate signal. When calculating the wavelet transform of this signal, the fundamental frequency will be the lowest (and strongest) frequency component, at 1 Hz. Since the wavelet transform shows the frequency across time, this 1 Hz component will show up spread across time, appearing as a consistent horizontal line. We say “low frequency horizontal line” because, since it is the fundamental frequency, it is at the lowest frequency clearly present in the wavelet transform.

The wavelet transform can reveal the bursting periodicity of the collective network firing. Having high spike count in a given time block says nothing about the regularity of these spikes across time. The firing rate can be high due to powerful bursts that engage the entire network but, if these bursts do not occur at periodically, then the wavelet transform will not reveal a clear frequency component across time. On the other hand, if these occur at a pace of 1 burst per second, for example, the wavelet transform will show the well-defined 1 Hz component at its base. We added the following clarification to the manuscript:

Page 5: “*Consistent inter-burst intervals (i.e., consistent neuronal oscillations) will appear as an horizontal line in the wavelet domain, corresponding to the fundamental frequency of the oscillation.”*

Figure 2:I missed seeing the same analysis performed in the random stimulation condition – it could be included as a supporting figure just for completeness.

We have now included it as a supporting Figure 3—figure supplement 2. We do not show it in Figure 2 because the main goal of this figure was to show that, in biological neurons (as opposed to computational models), conventional DFC may lead to worse oscillatory behaviour, which justifies the need for an adaptive algorithm. Here, we purposely present only qualitative analysis, leaving the quantitative results for the next sections, where we also include the comparison with random stimulation.

Section 2.2:The definition of firing rate should appear before, since it is one of the main variables used in the previous section.

As mentioned in a previous answer, we use both the notions of instantaneous firing rate and average firing rate. In the mentioned previous section of the manuscript, we refer to the instantaneous network firing rate, whereas in Section 2.2 we refer to the average firing rate over an entire 5 min block. We understand this difference may be overlooked by the reader, given the similarity of the names, and further clarified this distinction:

Page 6: “Synchrony measures the degree of co-activation of multiple electrodes (see Methods); firing rate corresponds to the average number of spikes per electrode per unit of time (not to be confused with the instantaneous network firing rate signal presented in Figures 2.C and D);”

Defining the oscillation frequency and intensity based on the main peak does not seem to make sense for the aDFC condition. Your premise is that this treatment disrupts periodicity (which it appears to do) then, in theory, there should not be a defined "peak" in the frequency domain. This identified peak will rest within the noise levels, rendering it "random". Maybe a threshold could be set, i.e., if the signal-to-noise ratio of a peak is X, then it is considered, otherwise the frequency and intensity of the oscillation are respectively none and zero. Furthermore, the "randomness" of this oscillation frequency can be seen in Figure 3, where the trials with aDFC show a very large range of main frequencies.

After inspecting in detail all the FFTs obtained for all the experiments performed, we noted that some of the results obtained with aDFC and Poisson were indeed capturing noise, as well-pointed by Reviewer #1. Following the suggestion, we decided to upgrade the analysis of the frequency domain, so that it takes into account the background noise. This has led to some important modifications in the manuscript and has improved, in fact, several of our results. We thank Reviewer #1 for this important comment/suggestion. These modifications are described below:

Regarding the Oscillation Intensity, instead of using the magnitude of the peak of the FFT as metric, we calculated the Power Spectrum Density (PSD) and calculated the signal-to-noise ratio (SNR) of the fundamental frequency. This method is more robust since it takes into account how the power of the peak compares with the power of the background noise. Also, the SNR is calculated assuring that the DC component and harmonics are not mistaken as background noise (MATLAB function **snr**). Therefore, the new metric of Oscillation Intensity is the difference between the SNR of the ON period (under stimulation) and the OFF period (before stimulation). Note that for the metrics Firing Rate and Synchrony, we calculated the ratio between the ON and OFF metrics, whereas here we calculate the difference. That is simply because the SNR is expressed in logarithmic scale, and so, to compare the evolution from OFF to ON we must calculate the difference. We updated the manuscript accordingly:

Page 6: “To quantify this effect, we calculated the magnitude of the main oscillation by calculating the signal-to-noise ratio (SNR) of the Power Spectrum Density (PSD) of the instantaneous firing rate for the ON and OFF segments (Figure 2.D and F).”

Page 7: “Their effects on a given network are evidenced by normalizing the values of each feature to the initial OFF period (before stimulation) across the multiple trials (Figure 3.A). This also reveals how the dynamics are recovered once the stimulation is turned OFF. Note that, for the Oscillation Intensity, this normalization corresponds to the difference between the ON and OFF periods, since the scale is logarithmic.”

Page 19: “The oscillation intensity is measured by calculating SNR of fundamental frequency of the Power Spectrum Density. This was calculated using the Matlab functions pwelch for the PSD and snr for the SNR, which computes the ratio of the summed squared magnitude of the fundamental frequency to that of the noise. The snr function automatically detects the fundamental frequency and assures that the harmonics and DC component are not mistaken as noise. The fundamental frequency corresponds to the frequency at the peak of the PSD.

For the firing rate and synchrony, the effect of each stimulation protocol was evaluated by calculating the metric’s fold change form OFF to ON. For the oscillation intensity we calculated the difference between the OFF and ON periods, since it is measured in dB, which is a logarithmic scale.”

Regarding the Oscillation Frequency metric, it is indeed problematic to attribute a value that could be just noise. Reviewer #1 suggested establishing a threshold for the SNR of the peak and considering only the cases where the SNR is above the threshold. However, we were afraid that the value chosen for the threshold could be quite arbitrary. Consequently, we decided to remove this problematic metric from the main analysis. This means that the definition of Controllable Network is no longer based on four metrics, but only the remaining three: Firing Rate, Synchrony and (the updated and more robust) Oscillation Intensity. Using these three metrics led to slight changes in Figures 3 and 4 (one network went from Controllable to Uncontrollable, and another from Uncontrollable to Controllable), but made no significant difference in the overall conclusions. Nonetheless, it is still interesting to comment on the alteration of the Oscillation Frequency (the frequency at the PSD peak), particularly when using the standard DFC, where it tends to increase. Following this, we included another figure in Supplementary Materials, Figure 3—figure supplement 3, where we plotted the change in Oscillation Frequency as a function of the SNR difference for all experiments. Our goal was to show that, at least for DFC, an increase in the SNR of the oscillation is followed by an increase in the oscillation frequency.

We thank again Reviewer #1 for this important comment which improved the clarity and the quality of our results. Re-analysing this data, and doing again several figures, was the reason why the submission of this revised version took longer. We would also like to state that, in the process of thoroughly revising these calculations and preparing the dataset for Zenodo, we found a couple of misplaced files in the original analysis (from a universe of 237 files). In the previous version, one of the uncontrollable networks in Figure 3E had three experiments (datapoints) that belonged to another network. After removing these three datapoints, this network no longer obeyed the criteria of having at least three experiments performed with each protocol (as stated in our Methods section). Consequently, we had to remove this uncontrollable network from the Figure 3. Since it was an uncontrollable network, it did not affect the results from Figure 4 (nor had any impact on the other results/conclusions of the paper). We also noticed that three files from the network in Figure 7 had not been included in the analysis, corresponding to two experiments with Poisson stimulation and one experiment with aDFC. Adding these experiments strengthened the conclusions, in fact, as the difference between aDFC and Poisson became even clearer.

Figure 3:How is the z-score calculated?

The z-score normalization centres the data to have mean 0, and scales it to have standard deviation of 1.

Legend of Figure 3E was updated to: “[…] The black traced line corresponds to the first principal component (PC1) of the standardized (z-score, i.e., mean equals zero and standard deviation equals one) synchrony and firing rate defining a relevant descriptor of the neuronal dynamics.”

Section 2.3:Figure 4A show oscillation intensity yet, in the main text, you claim that "standard DFC led to more consistent oscillations". How does intensity relate to consistency? DFC shows a higher inter-trial variance than the other conditions (less consistent?)

By “consistent oscillations”, we mean that the oscillation frequency becomes very stable during a given experiment with DFC (regardless of whether or not the frequency value itself is consistent across experiments). A very consistent oscillation periodicity will lead to a peak with high power in the Power Spectrum Density. We are now referring to SNR of the PSD peak as the Oscillation Intensity because it relates to how much the overall signal is dominated by the oscillatory component (note that before, we were evaluating the Oscillation Intensity by the amplitude of the peak of the Fourier Transform, but changed to the SNR of the peak because it is more robust and takes into account the background noise).

Indeed, DFC presents high inter-trial variance in the Oscillation Intensity (which is smaller now it the SNR measure) but, when we mentioned “standard DFC led to more consistent oscillations”, we were referring to the fact that there is an increase in the oscillation intensity. This entire paragraph re-written to clarify this point and to account for the fact that we updated the metrics and the statistical tests. It now reads:

Page 9: “The performances of DFC, aDFC and Poisson stimulation were evaluated by comparing their average effect on the three metrics across all the controllable network (Figure 4), where each datapoint corresponds to the metric’s mean for all the experiments performed with a given network and stimulation protocol (in Figure 3—figure supplement 2 we show all the datapoints of network, from which the means are extracted). Uncontrollable networks are not considered here as their results are, by definition, not reproducible. We included trials without stimulation for each network as a control group to capture the natural variability of each metric; in these, we calculated the metrics for two consecutive 5 min blocks of spontaneous activity, representing the initial OFF and ON periods.

The most significant distinction between control algorithms was evidenced at the level of the bursting oscillations (Figure 4.A). Both aDFC and random stimulation led to a significant decrease in Oscillation Intensity. The standard DFC algorithm, on the other hand, led to a significant increase in Oscillation Magnitude. This increase in magnitude was followed by an increase in the oscillation frequency (Figure 2A-C and Figure 3—figure supplement 3). The only stimulation protocol that led to a significant decrease in synchrony was aDFC (Figure 4.B) and its effect was significantly different form that of random stimulation. On the other hand, it was also the only method that significantly increased the average firing rate (Figure 4.D), even though the average stimulation frequency was similar for the three methods (Figure 3—figure supplement 3).”

Figure 4:All variables are studied in four different conditions, yet the authors use t-tests to evaluate statistical significance. This is the wrong statistical test to use – an ANOVA should be used. In this situation, multiple t-tests will lead to an increase in false positives.

We repeated the tests using ANOVA and updated the results in the manuscript. This did not affect the results/conclusions of the work. The changes made in the manuscript are already mentioned in the previous answer. We also updated the supplementary table with details of the statistical test. We thank Reviewer #1 for this correction.

Section 2.4:Did you quantify the excitatory/inhibitory balance in your MEA cultures? Could be interesting to compare to the simulation results.

We did not have access to the excitatory/inhibitory balance in the in vitro experiments, but it is an interesting point to consider in future experiments.

Section 2.5:The authors state that multiple trials were conducted in the same network but do not mention how long the network was allowed to recover in between trials, which is necessary information.

have been important to precisely control the period between experiments if the cultures took a long time to recover from the previous stimulation period. However, we saw that, once stimulation stopped, the firing dynamics returned to the baseline regime very quickly (Figure 2.A and B and Figure 3.A). We have added the following text to subsection 5.5 in the Methods:

Page 22, Methods: “The time interval between consecutive trials was in the order of 5 minutes. A precise time interval between trials was not required/imposed as, once stimulation stopped, the firing dynamics returned to the baseline regime very quickly (Figure 2.A and B and Figure 3.A).”

Furthermore, the authors state that only one network showed this effect (spontaneous AS time prior to stimulation). If only one network showed this among 14, how biologically relevant is this?

Indeed, from all the networks whose dynamics changed over time, only one showed brief and sporadic periods of spontaneous asynchrony consistently across the entire day of experiments. Still, the goal of this section is not on how biologically relevant (or frequent) this type of dynamics is but, instead, to show that aDFC could perform particularly well in this network that natively showed spontaneous transitions to brief periods of asynchrony. We hypothesize in the manuscript that the good performance could be due to the fact that this network could sustain asynchronous activity, while most networks cannot (at least with this minimally invasive stimulation protocol). In this situation, aDFC “just” have to promote the transition to the asynchronous activity state (already available), instead of creating and continuously forcing asynchronous activity. We believe this is an important idea, even if speculative.

This particular case also allows us to show that, once aDFC promoted the state transition, this state was maintained with almost no stimulation, a feature that is desirable for brain stimulation protocols. Also, this fact significantly differentiates aDFC from random Poisson which, as an open-loop protocol, thus does not adapt to the changing dynamics of the network. We expect this factor to be particularly relevant when moving to in vivo experiments, because brain networks are considerably more dynamic than in vitro networks and alternate between different synchrony states.

How can you be sure that this is not a consequence of a confounding variable (not properly developed network, etc). How old was this network?

We did not speculate on the reasons that the network displayed this multi-stability. Indeed, there may be many relevant variables governing the network dynamics, but it is unlikely that the network is underdeveloped because this experiment was performed on the 32 DIV and the network still exhibited abundant firing. However, the reasons behind the dynamics of such network were not the focus on this section, but rather to recognize that only aDFC forced the intended state transition in a network with such particular dynamics.

Was it also used in other DIVs?

Yes, this culture was used in both DIVs 32 and 37. This network is the one represented in blue in Figure 3—figure supplement 1.D. Its firing dynamics, and the way stimulation interfered with them, changed from day 32 to day 37. For example, in DIV 32 the network was multi-stable, and in DIV 37 it was not. Also the spontaneous dynamics occupy different regions of the spontaneous Synchrony vs Firing Rate map (Figure 3—figure supplement 1.D blue network).

Figure 7:The effect of the Poisson stimulation appears to be more similar to aDFC than random. A more in depth discussion would be helpful to better understand how a "random" stimulation could cause such a "block-type" effect.

Indeed, there are periods of asynchrony during random Poisson stimulation, but their prevalence is similar to what is found in spontaneous activity (Figure 7.G). So, it is not possible to say whether the asynchronous states were achieved due to the random Poisson stimulation or if they would still occur even without stimulation. On the other hand, with aDFC, the time spent in asynchronous state increased significantly.

Still, it is possible that a properly tuned random stimulation (as the one used here, regarding the frequency parameters of the Poisson process) could induce an asynchronous state. Maybe not through a “block-type” effect, such as the depolarization block mechanism proposed to occur in high-frequency deep brain stimulation [1], but instead by simply perturbing the spontaneously sustained firing oscillation; perhaps through an analogous mechanism to the “disruption” hypothesis that has been proposed to explain the effects of deep brain stimulation [1]. The “therapeutic” effect of adding noise is not particular to this situation. It is a “snow globe effect”: the dynamics are so affected that noise can be beneficial to bring it closer to “physiological” state (better than a patterned stimulus that imposes different but still pathological dynamics). Independently of the discussion on the mechanism(s), it is important to emphasize that, unlike aDFC, random stimulation is open loop, meaning that it does not adjust to the changing firing dynamics, nor does it stop stimulating once the desired state is achieved.

We added the following to the manuscript:

Page 15, Discussion: “The adaptive version, aDFC, effectively disrupted the oscillations and decreased network synchrony. Interestingly, random stimulation, using similar stimulation frequency, was also able to achieve this, although without significantly decreasing synchronization. Electrical stimulation may be acting as a disrupter of the emergent network dynamics, as it has been proposed for high-frequency deep brain stimulation (Chiken et al. 2016).”

[new REF] Chiken, Satomi, and Atsushi Nambu. "Mechanism of deep brain stimulation: inhibition, excitation, or disruption?" The neuroscientist 22.3 (2016): 313-322.

The comparison between OFF-ON-OFF states refers to a comparison of the same object in different timepoints. Thus, t-tests are not the correct statistical test here, but rather a repeated measures ANOVA (or non-parametric equivalent) must be used.

The statistical tests were repeated with ANOVA. Consequently, Figure 7 was updated as well as the supplementary tables with details of the statistical tests. There were no differences in the conclusions. We thank Reviewer #1 for this correction.

Methods:The authors state that experiments were performed between 13 and 55 DIV. When considering cultured neuron networks, this range makes a gigantic difference in their behavior, since the culture will not be fully mature at DIV 13, and the neuronal health at DIV 55 will already have declined. Only cultures at equivalent maturity levels should be compared.

The goal of this paper was to test/assess control algorithms on networks that exhibited synchronous oscillations. We believe we would have a less convincing work if we showed aDFC working only on cultures with equivalent maturity levels: by covering a wider range of situations, we can claim that the quality of the control is more related to the properties of the algorithm and less related to the specificities of the mechanism giving rise to the synchronous oscillations (the target behaviour of the control). So, our criteria to decide on which networks to perform the experiments were the firing dynamics exhibited by the culture, regardless of the DIV. We were selective by only choosing networks that presented synchronous oscillations with consistent inter-bursts intervals, ranging between 1-5 seconds.

We do not exclude the possibility that aDFC (or DFC, or Poisson) performs better/worse at specific maturity ranges, but we believe we present a more solid claim by covering multiple maturity levels (given that the focus is on the capabilities of the algorithm for disrupting synchronous oscillations). Also, we did not see any correlation between DIV of the selected networks and controllability (see Figure 3—figure supplement 1.E).

Authors define "network as a neuronal culture in a given day in vitro". This needs to be clarified for statistical analysis. With this definition, does it mean that if two experiments are performed in the same neuronal culture but in two different days, they are considered two independent networks? While the network behavior might change throughout days in vitro (and network maturity), it is not possible to state that the behavior of the network is independent from its previous state. Extra care should be had when performing statistical analysis on those data – they should be handled as the same "object" at different timepoints, NOT independent networks.

As stated, we are considering the same culture in a different DIV to be a different network. We refer to “network” as the dynamic system that is being subject to control. And, indeed, in different DIVs, the same culture does behave like a different dynamical system in the sense that the spontaneous dynamics are very different, as well as its controllability. This can be seen in Figure 3—figure supplement 1.D. Please notice that this distinction between culture and network in the context of controlling neuronal networks is not uncommon (see, for example, [1]). In our study, from the perspective of evaluating the effect of the different control algorithms, the same culture at different DIVs (with changed circuitry, connectivity level, baseline activity intensity, etc..), provide distinct dynamical systems. The statistical analysis was improved following the previous comments from Reviewer #1.

[1] Wülfing, Jan M., et al. "Adaptive long-term control of biological neural networks with deep reinforcement learning." *Neurocomputing* 342 (2019): 66-74.

The method used to determine the main frequency and intensity of the oscillation ASSUMES that there is a peak frequency, which may not be true and lead to erroneous results and conclusions.

Reviewer #1 is correct, and this was already addressed in a previous answer.

Reviewer #2 (Recommendations for the authors):This was a clear, well-written study. I only have a few suggestions/comments to the authors:0) Can you discuss whether you expect brain networks to operate in uncontrollable vs. controllable regimes? Is there evidence that brain networks should be controllable?

Our definition of controllable/uncontrollable networks can also be applied to brain networks in the sense that stimulating a particular brain networks will either generate a reproducible neuromodulation effect or not. In that sense, when deep brain stimulation is effective, it must be targeting a controllable network since it is consistently modulating the neuronal activity.

Neurostimulation controllability will depend, of course, on the particular brain network/circuit but also on the stimulation system: a network may be controllable with an array of stimulating electrodes, and uncontrollable using a single stimulating electrode.

Another interesting question is whether or not the spontaneous dynamics of a given brain network predict its propensity for neuromodulation (analogously to what we saw in vitro and *in silico*). The identification of such correlation would enable a more meticulous planning of the placement of neural stimulators. But, as far as we know, there is no evidence for that in vivo.

1) Can you establish a correspondence between the controllable subspace in the in silico model in Figure 5 and the controllable network properties in Figure 3? For example, can you make Figure 5B left (change in synchrony as a function of synaptic weight vs. excitatory neurons) into a synchrony as a function of firing rate vs. synchrony plot? Then superimpose data from Figure 3E so that one can see whether the in silico network properties are similarly predictive of controllable vs. uncontrollable networks as the in vitro network properties. If there isn't a clear correspondence between the in silico and in vitro networks, please discuss.

We added a supplementary figure (Figure 5—figure supplement 1) analogous to Figure 3 for the *in silico* results, where the X and Y axis correspond to the level of spontaneous synchrony and firing rate, respectively. The in vitro data is in the same order of magnitude as the controllable region seen *in silico* (spontaneous synchrony between 10^-1^ and 10^0^ and spontaneous firing rate between 10^0^ and 10^1^). But it is important to mention that the purpose of the *in silico* tests was not to exactly reproduce the results seen in vitro, but rather to test if specific network parameters could restrict/enable neuronal modulation. Still, on this new figure (Figure 5—figure supplement 1, analogous to Figure 3) it is possible to see a correspondence between the *in silico* model and in vitro networks.

2) Could you either provide experimental data or a discussion of previous studies that have tested conventional high frequency stimulation (~100-120 Hz) and studied its effects on neural synchronization? How might you expect your aDFC results to compare to conventional high frequency stimulation?

High-frequency stimulation (HFS) has been shown to disrupt pathological synchronization in disorders such as Parkinson’s disease [1] and epilepsy [2]. The question of whether aDFC would be more efficient in disrupting pathological oscillations than a properly tuned HFS remains unanswered. What can be expected is the fact that, once the network is desynchronized, aDFC will no longer impose strong stimulation on the network (some stimulation may still occur due to fluctuations of the actuation signal, but these may be advantageous in preventing synchronization to emerge again); HFS, on the other hand, would continue to stimulate regardless of its effect on the network. Also, if aDFC has the same efficacy as HFS, it will achieve that with a much less “aggressive” approach, since the premise of DFC algorithms is to disrupt the oscillation using small perturbations [3].

[1] Eusebio, Alexandre, et al. "Deep brain stimulation can suppress pathological synchronisation in parkinsonian patients." *Journal of Neurology, Neurosurgery & Psychiatry* 82.5 (2011): 569-573.

[2] Yu, Tao, et al. "High-frequency stimulation of anterior nucleus of thalamus desynchronizes epileptic network in humans." *Brain* 141.9 (2018): 2631-2643.

[3] Pyragas, Kestutis. "Delayed feedback control of chaos." *Philosophical Transactions of the Royal Society A: Mathematical, Physical and Engineering Sciences* 364.1846 (2006): 2309-2334.

We added the following information to the Discussion:

Page 15: “Whether aDFC is more efficient in disrupting pathological oscillations than high-frequency stimulation (HFS) remains unknown but, as opposed to fixed/continuous HFS, aDFC stimulation adapts to the population oscillation frequency and ceases once the network is desynchronized.”

There is some mention in the discussion of a study that uses 10-50 Hz and is able to prevent synchronous states from forming at the cost of a high firing rate. Why is the high firing rate a problem? Might this be the source of undesirable side effects of stimulation? Please discuss this in more detail in the discussion as it directly sheds light on how useful your algorithm may be compared to conventional stimulation.

The pathological neuronal synchronization seen in many neurological disorders is typically associated with an increased firing rate. As such, an algorithm that disrupts the pathological oscillation at the cost of increasing the firing rate even further may not be ideal, as it may be pushing the network into another pathological state. In that study mentioned in the discussion [1] they noticed that, when the networks were forced to fire at a rate much higher than the baseline levels (from the spontaneous 20 spikes per second array-wide to 400 spikes per second), the networks could no longer sustain synchronized activity. This is an interesting result, but it could be argued that the pathological synchronization was replaced by another abnormal firing pattern. Our aDFC protocol also increased the firing rate of the network, but only by 16% (see Supplementary File 1 (Table S1)). As a side note, the average increase in firing rate for the other two algorithms was higher (33% and 23% for DFC and Poisson, respectively), although with low statistical significance. Still, a slight increase in firing rate is not surprising considering that we are providing an excitatory signal to the network. The ideal scenario where excitatory electrical stimulation can actually decrease the firing rate is when a closed-loop algorithm can force a transition to an asynchronous state and stops/reduces stimulation as a consequence of that, just like it happed for the multi-stable network under aDFC (Figure 7).

[1] Wagenaar, Daniel A., et al. "Controlling bursting in cortical cultures with closed-loop multi-electrode stimulation." *Journal of Neuroscience* 25.3 (2005): 680-688.

We added the following information to the Discussion:

Page 15: “Other methods can achieve network desynchronization by following approaches that are more aggressive. Namely, it was already shown that continuous high frequency stimulation (10-50 Hz), distributed across multiple electrodes, can prevent network synchronization (Wagenaar 2005). This is done, however, at the cost of highly increasing the overall network’s firing rate, thus possibly establishing yet another non-physiological state. The goal of avoiding an abnormal prolonged increase in the population’s firing rate is grounded on the understanding that excessive activity is also detrimental in a neuronal circuit.”

3) Please discuss in the discussion the limits of the in vitro model, particularly in reference to the low frequency synchrony that the populations exhibit (~1 hz) compared to the pathological oscillatory activity thought to be relevant to PD (15-30 hz) or ET (5-7 Hz). Do you expect your algorithm to have difficulties suppressing these higher frequency oscillations?

Our in vitro model exhibited oscillations that are slower than those occurring in neuronal disorders such as Parkinson’s Disease or Essential Tremor. But there is nothing in the DFC method itself forcing it to work only within certain frequency ranges. Please notice that, by construction, DFC responds directly to the system frequency, be it 1 Hz or 30 Hz. In fact, DFC has been already implemented in many different non-neurological substrates, with a with range of oscillation frequencies, such as electrochemical systems, electrical oscillators, mechanical pendulums, among others [1].

Having said that, it is possible that the faster pathological oscillations seen in vivo may be harder to disrupt, not necessarily for being faster, but because the mechanisms underlying the oscillations may be very different from those occurring in vitro. Nonetheless, it is important to note that the oscillations seen in the in vitro models are more prominent than those seen in vivo, which only last brief periods of time [2]. The more consistent in vitro oscillations could be due to the exacerbated neuronal connectivity that emerges in these unconstrained cultures. It is reasonable to admit that a network whose spontaneous activity consists only of synchronized firing poses a more difficult challenge for the aDFC algorithm (or any other) than a network that can naturally sustain different firing regimes, such as in vivo networks.

[1] Pyragas, Kestutis. "Delayed feedback control of chaos." *Philosophical Transactions of the Royal Society A: Mathematical, Physical and Engineering Sciences* 364.1846 (2006): 2309-2334.

[2] Tinkhauser, Gerd, et al. "The modulatory effect of adaptive deep brain stimulation on β bursts in Parkinson’s disease." *Brain* 140.4 (2017): 1053-1067.

Are there in vitro models that exhibit higher frequency oscillations that could be used to test this algorithm?

Dissociated neuronal cultures present this type of slower oscillations [3], but in vitro brain slices can display much faster oscillations, in the orders of dozens of Hz [4]. Organotypic brain slices can be cultured on MEAs and be viable for several weeks. This would be an interesting model to further test the algorithm, but we hope Reviewer #2 agrees that this would be another study by itself.

[3] Wagenaar, Daniel A., Jerome Pine, and Steve M. Potter. "An extremely rich repertoire of bursting patterns during the development of cortical cultures." *BMC Neurosc.* 7.1 (2006): 1-18.

[4] Dulla, Chris G., et al. "How do we use in vitro models to understand epileptiform and ictal activity? A report of the TASK 1‐WG 4 group of the ILAE/AES Joint Translational Task Force." *Epilepsia Open* 3.4 (2018): 460-473.

4) Can the authors comment on how many units must be monitored from in order to make an accurate estimate of the underlying neural population's oscillatory frequency?

Interesting question. If most of the recording electrodes are capturing synchronous oscillatory activity, then the (population) instantaneous firing rate will show clear periodic peaks from which the oscillation frequency can be accurately estimated (and, on the limit, a single recording unit would suffice). On the other hand, if only a small percentage of the recording electrodes capture synchronized firing and the rest capture sparse firing, then the instantaneous population firing rate may not show a clear periodic signal (and more recording units are required). Thus, it is not only a matter of how many recording units are available but also, how present the oscillation is compared with the sparse firing.

Can LFP features be used instead of multiple single neuron recordings? This has a direct bearing on how readily translatable this technology is to current neurostimulation devices.

The *in silico* studies use LFP as the tracked variable [1-3]. In our study, since we were working with dissociated cultures over MEAs, we had access to the extracellular spikes. That is why our DFC and aDFC algorithms are based on the instantaneous population firing rate signal, calculated from the spike data. But LFPs could have been used instead: analogously to the instantaneous population firing rate, LFPs will have a peak when the population fires in synchrony. Working with the LFP instead of individual spikes would actually make the workflow simpler, since this signal could be used directly to feed the DFC algorithms.

[1] Liu, Chen, et al. "Delayed feedback-based suppression of pathological oscillations in a neural mass model." *IEEE Transactions on Cybernetics* 51.10 (2019): 5046-5056.

[2] Daneshzand, Mohammad, Miad Faezipour, and Buket D. Barkana. "Robust desynchronization of Parkinson’s disease pathological oscillations by frequency modulation of delayed feedback deep brain stimulation." *PloS one* 13.11 (2018): e0207761.

[3] Popovych, Oleksandr V., Christian Hauptmann, and Peter A. Tass. "Control of neuronal synchrony by nonlinear delayed feedback." *Biological cybernetics* 95.1 (2006): 69-85.

5) During aDFC how often were the frequency (w) and period time (T) updated?

As described in the manuscript, the frequency and period are updated whenever the median of the last 5 inter-burst interval changes. The median is calculated when a new burst is detected, meaning that, in theory, the value of *w* and *T* can change every time a new burst arrives. However, for a drastic change in the values of *w* and *T* – for example, the population was firing consistently at 1Hz and suddenly changes to 2Hz – the algorithm requires three 1/(2Hz) = 0.5s intervals between bursts for the median T to be updated to 0.5s, and the w to 2Hs: [1, 1, **0.5**, 0.5, 0.5]s. This can be seen in the Figure 1—figure supplement 1.

6) Where was the ground for the stimulation electrode?

The experiments were performed on 9well and 6well MEAs. Each well has an independent culture with recording/stimulating electrodes and a ground electrode. The ground electrodes are placed as shown here:

9 well MEA: https://www.multichannelsystems.com/products/256-9wellmea30030ir-ito-mq

6 well MEA: https://www.multichannelsystems.com/products/256-6wellmea20030ir-ito-rcr

Does your stimulation better mimic bipolar or monopolar stimulation in current therapeutic DBS therapies?

Our stimulation is monopolar: the circuit is closed between each stimulating electrode and a common GND. Still, one needs to be careful on a direct comparison between in vivo DBS and in vitro stimulation, as conditions are different.

What was the impedance of the stimulation electrode?

For the used electrodes (titanium nitride, planar and with 30 µm diameter) the impedance is < 100 kΩ.

Some neuromodulation fields use current (not volts) to report their stimulation amplitude and it would be helpful for those building on this study to have both quantities reported. In addition, please report the electrode surface area so that current density calculations can be made.

As described, in this study stimulation is provided as voltage pulses. Stimulation amplitude, either I or V, is set at the level of robustly eliciting a neuronal response but still within physiological ranges. For in vitro cultures, V stim is typically in the order of 500-800 µV amplitude and duration of tenths of µs, whereas I stim amplitude is in the order of tenths of µA.

We added the following information to the Discussion:

Page 17: “Each well of the 6-well MEA has 42 TiN circular electrodes (array of 7 × 6), and the 9-well contain arrays of 26 TiN circular electrodes (6 x 5 without corners), each with a diameter of 30 µm.”

7) Why was stimulation frequency the metric that was updated by DFC and aDFC as opposed to stimulation amplitude?

We had two options: either keep the amplitude fixed and modulate the stimulation frequency, or keep a constant stimulation frequency and modulate the stimulus amplitude. We chose the former for two main reasons. First, it was easier to stablish a pertinent stimulation amplitude, following our protocol of setting the minimum amplitude that elicited a neuronal response. Fixing the stimulation frequency would require a somewhat arbitrary choice of stimulation frequency, or would force us to test different frequencies, which would unnecessarily expand the testing conditions. The second reason is practical: given the way the hardware and software were set up, it was easier to fix the amplitude and have the computer trigger the stimulus than to have a new stimulus waveform being download to the stimulus generator at every stimulation instance.